# Unveiling the Learning Mind of Language Models: A Cognitive Framework and Empirical Study

**Zhengyu Hu**[1,2], **Jianxun Lian**[3,*], **Zheyuan Xiao**[1,2], **Seraphina Zhang**[4],
**Tianfu Wang**[1,2], **Nicholas Jing Yuan**[5], **Xing Xie**[3], **Hui Xiong**[1,2]

[1]Thrust of Artificial Intelligence,
The Hong Kong University of Science and Technology (Guangzhou), China
[2]Department of Computer Science and Engineering,
The Hong Kong University of Science and Technology Hong Kong SAR, China
[3] Microsoft Research Asia    [4] University of Cambridge    [5] Microsoft.

## Abstract

Large language models (LLMs) have shown impressive capabilities across tasks such as mathematics, coding, and reasoning, yet their learning ability, which is crucial for adapting to dynamic environments and acquiring new knowledge, remains underexplored. In this work, we address this gap by introducing a framework inspired by cognitive psychology and education. Specifically, we decompose general learning ability into three distinct, complementary dimensions: *Learning from Instructor* (acquiring knowledge via explicit guidance), *Learning from Concept* (internalizing abstract structures and generalizing to new contexts), and *Learning from Experience* (adapting through accumulated exploration and feedback). We conduct a comprehensive empirical study across the three learning dimensions and identify several insightful findings, such as (i) interaction improves learning; (ii) conceptual understanding is scale-emergent and benefits larger models; and (iii) LLMs are effective few-shot learners but not many-shot learners. Based on our framework and empirical findings, we introduce a benchmark that provides a unified and realistic evaluation of LLMs' general learning abilities across three learning cognition dimensions. It enables diagnostic insights and supports evaluation and development of more adaptive and human-like models. The code is available here [2].

## 1   Introduction

Large language models (LLMs) [3, 2, 88, 46] have demonstrated impressive capabilities across diverse tasks such as mathematics [97, 94, 92], coding [60, 11, 39], and reasoning [29, 90, 99, 50]. Despite these achievements, as LLM-driven systems increasingly integrate into societal roles, serving as AI companions [87, 8, 64] or AI employees [6, 53, 75, 20, 37], they must possess robust learning capabilities. Effective learning is essential for AI agents to adapt dynamically to diverse and changing environments, continuously acquire new knowledge, and autonomously respond to novel contexts. However, research on the learning ability of LLMs remains scarce, with little work systematically investigating how well LLMs can acquire and generalize new knowledge across tasks.

To address this gap, we aim to systematically investigate the learning ability of LLMs. Drawing insights from cognitive science and educational theory [24, 42, 14], we first analogize from human learning processes to identify three fundamental dimensions of learning: (1) learning from instructor, where the model acquires knowledge through guided interaction [18, 72]; (2) learning from concept,

---

*Corresponding author
[2]`https://aka.ms/learnarena`

39th Conference on Neural Information Processing Systems (NeurIPS 2025).

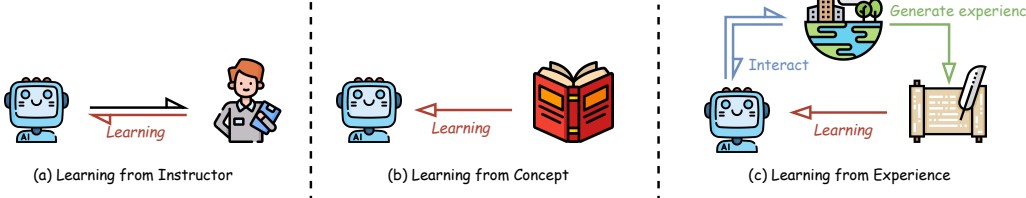

(a) Learning from Instructor     (b) Learning from Concept     (c) Learning from Experience

Figure 1: Overview of our proposed cognitive framework for evaluating general learning abilities in LLMs. We decompose learning into three core types: (a) Learning from Instructor; (b) Learning from Concept; and (c) Learning from Experience.

where the model internalizes structured abstractions and generalizes them to downstream tasks [56, 5]; and (3) learning from experience, where the model adapts based on accumulated trajectories or exploration-feedback [42, 98]. For each dimension, we design targeted experimental paradigms to operationalize the corresponding learning mechanism. In Learning from Instructor, we simulate tutor-learner settings with and without interactive clarification, demonstrating that interaction within instructor and learner consistently improves model's learning ability. In Learning from Concept, we evaluate the impact of injecting abstract conceptual knowledge in competitive environments (i.e., TextArena [28]), showing that (i) conceptual understanding is scale-emergent, and (ii) injecting structured domain knowledge can provide a tangible advantage, if the model is sufficiently capable of internalizing it. In Learning from Experience, it is a crucial capability for adapting to novel environments and acquiring new knowledge autonomously. While they are few-shot learners, they struggle in many-shot settings due to the challenges of long-context integration. This highlights the importance of a unified benchmark that can evaluate LLMs' general learning abilities across cognitive dimensions.

Building on this framework and empirical insights, we consolidate our findings into a unified benchmark, LearnArena, that reflects realistic and cognitively grounded learning scenarios. It enables principled evaluation of LLMs' learning behavior across three learning aspects, and provides a foundation for advancing learning capabilities. Empirical results reveal that learning ability benefits from increased capacity, but faces a bottleneck; architectural and training advancements play a crucial role in further enhancing learning capability.

Our contributions are as follows: (1) We present the first work to explicitly evaluate and analyze the general learning ability of LLMs. Grounded in cognitive science, we propose a principled decomposition into three dimensions: learning from instructor, learning from concept, and learning from experience, each with dedicated methodologies. (2) We conduct a comprehensive empirical study across the three learning dimensions, revealing three key insights: interaction improves learning in instructor-based settings; conceptual understanding is scale-emergent and benefits larger models; and LLMs are effective few-shot learners but not many-shot learners. (3) Based on our framework and findings, we introduce a benchmark, LearnArena, that offers a unified and realistic evaluation of LLMs' general learning ability across three cognitive dimensions. It enables diagnostic insights and supports the development of more adaptive and human-like models.

## 2 Related Work

**Evaluation of Large Language Models.** LLMs have been extensively evaluated on tasks involving linguistic competence [77, 9, 40], factual recall [27, 81], reasoning [49, 51, 90, 99, 96, 95], instruction following [35, 45, 36, 44, 34], and multitask generalization [59, 17, 89, 33], with benchmarks such as MMLU [30], BIG-Bench [1], and HELM [47] highlighting broad capabilities and emergent scaling trends. However, most evaluations focus on static zero- or few-shot performance, offering limited insight into how models learn or generalize from experience. To address this, recent studies have begun shifting toward more dynamic formulations that probe the learning behavior of LLMs. [21] categorize in-context learning (ICL) as inference-time adaptation and identify key factors for success. [55] show that ICL performance depends more on input format than label correctness. [4] find that many-shot ICL yields diminishing returns as context length grows. [68] introduce Toolformer, enabling models to self-supervise API use. [91] develop models that improve via internally generated feedback. Nevertheless, most such studies are piecemeal, lacking a unified theoretical lens or

systematic analysis for comparing learning processes across tasks. Our work addresses this gap by introducing a cognitively grounded decomposition to systematically analyze learning ability.

**Learning Ability of Large Language Models.** Recent advances in large language models have demonstrated remarkable capabilities in adapting and generalizing knowledge beyond traditional static evaluations [7, 84, 78]. Cognitive psychology and educational theory have long emphasized that effective learning involves not only direct instruction and guided feedback [14, 18], but also the ability to abstract conceptual structures and adapt from experience [42, 56, 24]. Motivated by these insights, we adopt a tripartite perspective, learning from instructor, concept, and experience, that mirrors rapid-instruction-to-implementation learning (RITL) in humans [72], classical rule-based accounts of cognition [56, 24], and experiential learning theory [42]. While contemporary LLM benchmarks extensively measure task-specific performance [1, 47, 30, 54], they rarely probe how models internalize instructions, extract explicit rules, or accumulate and reuse episodic experience, capacities increasingly highlighted by self-evolving or self-refinement agents [26, 32]. This gap underscores the need for a cognitively informed evaluation framework that systematically characterizes the general learning abilities of LLMs across these complementary dimensions.

## 3 A Cognitive Framework for Analyzing Learning Abilities in LLMs

To systematically investigate the general learning capabilities of LLMs, we propose a cognitively grounded framework comprising three paradigms: *Learning from Instructor (LfI)*, *Learning from Concept (LfC)*, and *Learning from Experience (LfE)*. LfI captures learning via explicit guidance, supported by evidence that instruction accelerates task acquisition and reduces cognitive load [18, 72, 14, 76]. LfC reflects structured abstraction from static concepts, grounded in symbolic cognitive theories [56, 24, 32] despite critiques of their rigidity [5]. LfE models adaptation through feedback and interaction, consistent with experiential learning theory [42, 38] and recent work showing LLMs improve via self-refinement [98, 26].

Formally, we define a learning task $t$ as a tuple $(\mathcal{X}_t, \mathcal{Y}_t, \mathcal{C}_t)$, where $\mathcal{X}_t$ denotes the input space, $\mathcal{Y}_t$ the output space, and $\mathcal{C}_t$ the context space, which characterizes the type of information the model must incorporate to generate correct outputs. The learning objective is to establish a predictive mapping:

$$f_\theta : \mathcal{X}_t \times \mathcal{C}_t \to \mathcal{Y}_t \qquad (1)$$

where the form of $\mathcal{C}_t$ is determined by the learning paradigm.

In **Learning from Instructor (LfI)**, the model acquires task knowledge through explicit interactions with an external instructor, which may involve demonstrations, explanations, or corrections. These interactions form a communication-based supervision channel, where the instructor incrementally shapes the model's understanding of the task. Formally, we define the context as $\mathcal{C}_t = \mathcal{I}_t$, where $\mathcal{I}_t$ encodes structured instructional signals such as natural language directives, step-by-step exemplars, or annotated feedback. The model learns to integrate these interactive signals into its prediction process: $f_\theta(x, \mathcal{I}_t) \to \hat{y}$.

In **Learning from Concept (LfC)**, the model receives static, abstract conceptual knowledge that captures abstract properties or domain principles. Unlike instructor signals that evolve through interaction, concepts are predefined and non-interactive, such as definitions, category structures, or rule schemata. We define the context as $\mathcal{C}_t = \mathcal{K}_t$, where $\mathcal{K}_t$ represents a set of symbolic or linguistic descriptions encoding domain-relevant concepts. The model is expected to internalize these abstractions and apply them to generate predictions consistent with conceptual constraints: $f_\theta(x, \mathcal{K}_t) \to \hat{y}$, subject to $\hat{y}$ being semantically aligned with $\mathcal{K}_t$.

In **Learning from Experience (LfE)**, the model accumulates and utilizes its own prior interaction history to adapt future behavior. Unlike LfC, where supervision is static and predefined, LfI and LfE both involve dynamic supervision. In LfI, the model receives guidance through interactive instructions, while in LfE, supervision emerges from sequences of interactional feedback, potentially including successful or failed trajectories, user preferences, or latent task rewards. The context is defined as $\mathcal{C}_t = \tau_t = \{s_1, s_2, \ldots, s_k\}$, where each $s_i$ represents a structured snapshot from a past interaction, such as input-output pairs, action-state transitions, or dialog turns. The model is expected to generalize from this accumulated experience to improve decision-making: $f_\theta(x, \tau_t) \to \hat{y}$.

These three paradigms offer a unified and cognitively motivated framework for evaluating how LLMs acquire, organize, and apply knowledge. LfI emphasizes guided learning via instructor interaction;

LfC emphasizes structural abstraction from fixed knowledge; and LfE emphasizes adaptation through situated experience. This decomposition allows us to probe distinct facets of model generalization and align LLM evaluation with core dimensions of human learning.

# 4 Learning from Instructor (LfI)

We investigate Learning from Instructor (LfI), where models acquire task knowledge via structured guidance, such as demonstrations, explanations, or feedback. We evaluate this setting across two dimensions: (1) *Passive Consumption vs. Interactive Clarification*, and (2) *Scaling Learner*.

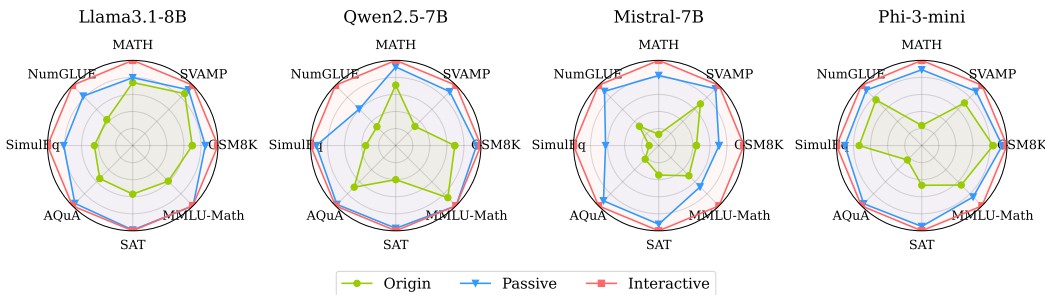

Figure 2: Comparison of learner performance under *Passive Consumption* and *Interactive Clarification* paradigms across eight mathematical benchmarks.

## 4.1 Passive Consumption vs. Interactive Clarification

**Experiment Setup.** To examine the impact of interactivity in instruction-based learning, we adopt the MagpieMath [86] dataset, a high-quality math dataset generated autoregressively by Qwen2.5-Math-72B. An instructor model is first trained on this dataset and subsequently used to teach a separate learner model under two learning paradigms: (1) Passive Consumption, in which the learner receives only direct solutions from the instructor without any further interaction, and (2) Interactive Clarification, where the learner is allowed to ask clarification questions following each instructor response, and the instructor provides targeted, follow-up explanations. In this setting, the learner is restricted to a single clarification question per sample to ensure consistency across training examples. Both the initial answers and the clarification responses are aggregated as supervision for the learner. We experiment with four representative LLM families: LLaMA3.1-8B, Qwen2.5-7B, Mistral-7B, and Phi-3-mini—each used as both instructor and learner to test robustness across pairings. Learners are trained on outputs from matched instructors within each setting. Evaluation is conducted on eight diverse mathematical benchmarks: GSM8K [16], SVAMP [63], MATH [31], NumGLUE [57], SimulEq [43], AQuA [48], SAT [101], and MMLU-Math [30], covering competencies from basic arithmetic to multi-step symbolic reasoning and standardized test preparation. Implementation and generation details are provided in Appendix A.3.

**Main Result.** As shown in Figure 2, learners trained under the *Interactive Clarification* paradigm consistently outperform those trained via *Passive Consumption* across all eight evaluation benchmarks. This performance gain highlights that LLMs are capable of leveraging interactive feedback to improve task understanding, resembling human-like active learning behaviors. Notably, the gains vary across model families. Mid-sized models such as Mistral-7B and Qwen2.5-7B show substantial improvements, suggesting a strong capacity to benefit from additional instructional signals. In contrast, the smallest model, Phi-3-mini, shows only marginal improvement, indicating limited ability to engage in or benefit from interactive clarification, which suggests that active learning capabilities are not uniform across models and may depend on model capacity or architecture.

## 4.2 Scaling Learner

**Experiment Setup.** While Section 4.1 explores the effect of instructional interactivity across different model families, here we focus on how scaling the *learner* model within the same family influences learning outcomes. Specifically, we fix the instructor as Qwen2.5-72B and vary the learner

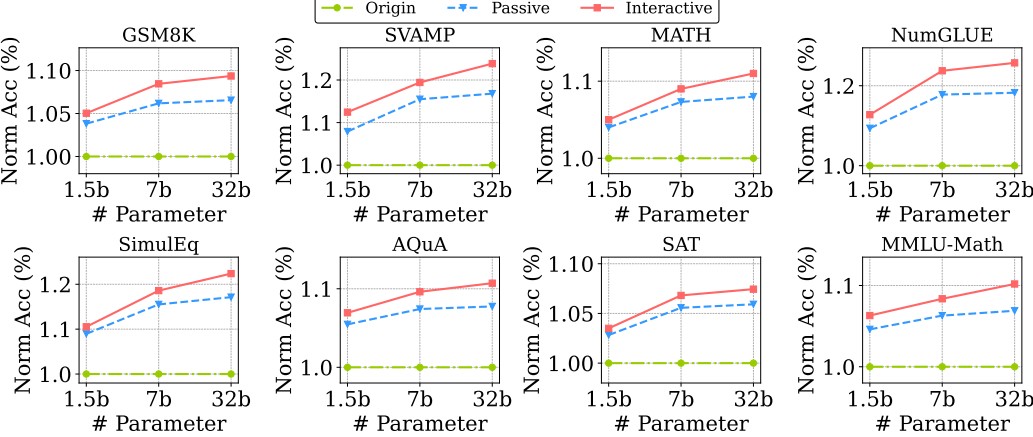

Figure 3: Scaling learner models (Qwen2.5 variants) with a fixed Qwen2.5-72B instructor. Larger learners learn more effectively, and interactive clarification further improves outcomes.

model across Qwen2.5-1.5B, Qwen2.5-7B, and Qwen2.5-32B. We evaluate the impact of instructional interactivity by comparing the *Passive Consumption* and *Interactive Clarification* paradigms. Training data and evaluation benchmarks are kept consistent with Section 4.1. Results are shown in Figure 3, where Origin refers to the performance of the untrained model, and Norm Acc denotes the accuracy of the post-training under each setting divided by its corresponding untrained baseline. Further implementation details are provided in Appendix A.3.

**Main Result.** As shown in Figure 3, we observe a strong positive correlation between learner model scale and normalized learning gains: larger models consistently achieve higher performance improvements across all evaluation tasks. This trend holds for both *Passive Consumption* and *Interactive Clarification*, with the benefits of interactivity becoming more pronounced as model capacity increases. Qwen2.5-32B demonstrates substantial gains under interactive supervision across all eight math-related benchmarks. For instance, in NUMGLUE and SVAMP, it achieves +25.7% and +23.8% normalized improvement, compared to +18.2% and +16.8% in the passive setting—showing clear added value from clarification feedback. Similar trends appear in SIMULEQ and MATH, where interactive training leads to +22.4% and +11.0% improvements, respectively. In contrast, Qwen2.5-1.5B shows smaller gains overall, and the gap between passive and interactive learning is narrower. For example, in GSM8K and SAT, improvements under interactivity are only +5.0% and +3.5%, compared to +3.8% and +2.8% in the passive case. This indicates that limited capacity restricts the ability to absorb and act on richer supervision signals.

### 4.3 Scale Instructor

**Experiment Setup.** We examine how the scale of the *instructor model* influences learning effectiveness under different supervision paradigms. Specifically, we fix the learner model as Qwen2.5-7B and vary the instructor across Qwen2.5-1.5B, 7B, 32B, and 72B. In each setting, we compare two learning paradigms: *Passive Consumption*, where the learner receives static demonstrations, and *Interactive Clarification*, where the learner can ask follow-up questions. The training setup and evaluation benchmarks follow those in Section 4.1, and additional implementation details are provided in Appendix A.3.

**Main Result.** However, this trend breaks when the instructor model is extremely small (e.g., Qwen2.5-1.5B). In such cases, interactive clarification may introduce noise or misleading guidance, leading to degraded performance compared to passive learning. For example, on tasks such as GSM8K, MATH, and SAT in Figure 4, the interactive setting underperforms the passive counterpart, suggesting that low-quality instructors are unable to generate helpful responses to clarification queries, which in turn hinders learner improvement. As shown in Figure 4, larger instructor models consistently lead to better learning outcomes. When the instructor is strong (e.g., Qwen2.5-32B or 72B), interactive supervision significantly improves learner performance compared to passive

instruction. This suggests that powerful instructors can provide useful clarifications that facilitate deeper understanding and generalization. However, when the instructor is too small (e.g., Qwen2.5-1.5B), the benefits of interactivity diminish or even become negative. In tasks such as GSM8K, MATH, and SAT, we observe that learners trained with *Interactive Clarification* underperform compared to those trained passively. This indicates that weak instructors may introduce noise or incorrect signals during interaction, ultimately hindering learning. These findings highlight the importance of instructor quality in interactive learning and suggest that poorly performing instructors can negatively affect the training process.

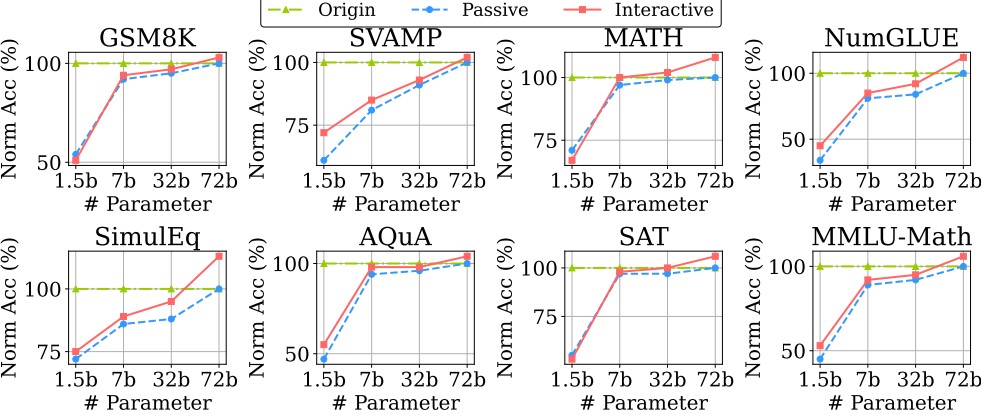

Figure 4: Scaling instructor models (Qwen2.5 variants) with a fixed Qwen2.5-7B learner. Weak instructors can degrade learning in the interactive setting; however, interaction becomes beneficial as instructor capability increases.

## 5 Learning from Concept (LfC)

We study Learning from Concept (LfC), where models leverage static, abstract knowledge, such as rules, definitions, or structured representations, to guide behavior or reasoning. We evaluate this ability in two settings: (1) *Structured Knowledge Injection in Competitive Environments*, testing whether models can integrate conceptual hints to improve decision-making in competitive environments, and (2) *Conceptual Generalization in Logic and Planning Tasks*, examining generalization from symbolic structures in logic and planning tasks.

### 5.1 Structured Knowledge Injection in Competitive Environments

**Experiment Setup.** We evaluate whether conceptual knowledge improves strategic performance in multi-agent settings using TextArena [28], a suite of competitive environments with symbolic rules and multi-turn dynamics. Each game involves two players, Player-0 and Player-1. We fix Player-0 as Qwen2.5-32B and vary Player-1 across four scales: Qwen2.5-1.5B, 7B, 14B, and 32B. Prior to gameplay, Player-1 either receives no guidance (*without Concept*) or is given natural language descriptions of game rules and strategies (*with Concept*), generated by Qwen2.5-32B. For each environment and model pair, we run 20 matches and report Player-1's win rate averaged over these games. Environments include Checkers (CH), Poker (PK), Stratego (ST), Tic Tac Toe (TT), Truth and Deception (TD), and Ultimate Tic Tac Toe (UTT). Performance is measured as Player-1's win rate against the fixed opponent. Results are shown in Figure 5. Implementation details are provided in Appendix A.4.

**Main Result.** Results are shown in Figure 5. We observe two key trends. First, model scale significantly influences the effectiveness of conceptual guidance. For smaller models such as Qwen2.5-1.5B, injecting conceptual descriptions consistently degrades performance across all environments—*with Concept* underperforms *without Concept*, suggesting that low-capacity models struggle to integrate abstract knowledge and may treat it as distractive noise. In contrast, as model size increases, the gap progressively narrows and eventually reverses. By Qwen2.5-14B and 32B, models consistently benefit from conceptual input across most tasks, indicating improved abstraction and utilization

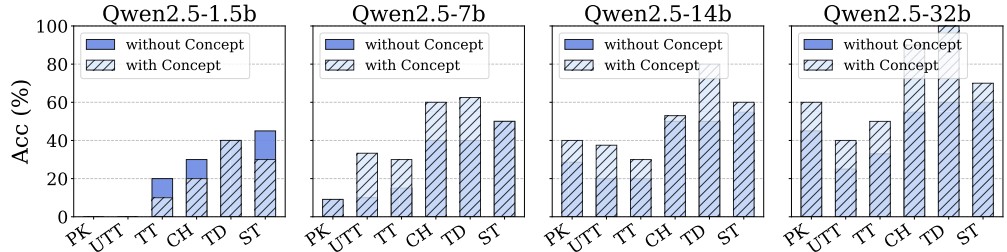

Figure 5: Win rates of Player-1 across six competitive environments from TextArena under the LfC setting.

capabilities with scale. Second, win rates across tasks also increase monotonically with model size, regardless of condition, highlighting a strong correlation between scale and general strategic competence. Notably, high-conceptual-load environments such as Stratego and Truth and Deception show the largest gains from conceptual input, especially for larger models. These results suggest that (i) conceptual understanding is scale-emergent, and (ii) injecting structured domain knowledge can provide a tangible advantage, if the model is sufficiently capable of internalizing it.

## 5.2 Conceptual Generalization in Logic and Planning Tasks

**Experiment Setup.** To evaluate rule-based generalization, we include six tasks centered on explicit symbolic or logical structures. LogicGrid and NaLogic feature structured and narrative logic puzzles [52, 23], testing deduction under constraint. Plan [79] translates classical symbolic planning domains (e.g., Gripper, Barman) into natural language settings to assess structured action reasoning. Additionally, tasks from AlfWorld [74], ScienceWorld [82], and BabyAI [10], models must complete goal-oriented tasks requiring commonsense reasoning, procedural knowledge, and spatial navigation. For the concept-injected setting, we use Qwen2.5-72B to generate static conceptual hints, which are provided as auxiliary input to smaller models during evaluation. Results are presented in Table 1 and Figure 5. We use subgoal-based annotations to enable fine-grained progress tracking. Implementation details are provided in Appendix A.4.

Table 1: Accuracy on six tasks with and without concept input (baseline / +concept). Higher scores indicate better subgoal completion or reasoning accuracy. Darker cell colors indicate larger improvements from concept input; white means no change.

| Qwen 2.5 | 7b | 14b | 32b | 72b |
|---|---|---|---|---|
| LogicGrid | 0.36/0.42 | 0.51/0.54 | 0.53/0.56 | 0.54/0.57 |
| NaLogic | 0.13/0.16 | 0.14/0.18 | 0.17/0.20 | 0.21/0.23 |
| Plan | 0.14/0.17 | 0.32/0.34 | 0.33/0.34 | 0.33/0.35 |
| AlfWorld | 0.27/0.29 | 0.30/0.32 | 0.33/0.34 | 0.33/0.33 |
| ScienceWorld | 0.16/0.19 | 0.26/0.29 | 0.33/0.35 | 0.41/0.42 |
| BabyAI | 0.40/0.42 | 0.41/0.44 | 0.46/0.46 | 0.51/0.51 |

**Main Result.** As shown in Table 1, providing external conceptual information, generated by Qwen2.5-72B, consistently improves performance across all model sizes and tasks. The improvements are most pronounced in smaller models, indicating that such auxiliary knowledge can partially offset limited inherent reasoning capacity. In symbolic reasoning tasks such as LogicGrid and NaLogic, concept injection yields consistent gains. For example, LogicGrid accuracy increases from 0.36 to 0.42 for Qwen2.5-7B, and from 0.54 to 0.57 for Qwen2.5-72B, suggesting that explicit structural cues remain beneficial even for larger models, though with reduced marginal benefit. In planning tasks like Plan, the gains are smaller but stable, e.g., from 0.14 to 0.17 in Qwen2.5-7B and 0.32 to 0.34 in Qwen2.5-14B, indicating that models can utilize conceptual breakdowns of action dynamics, particularly when capacity is constrained. For interactive environments such as AlfWorld, ScienceWorld, and BabyAI, we observe moderate but limited improvements. While concept injection provides useful structural cues, these tasks involve complex dynamics, such as exploration, multi-step control, and environment grounding, that are not fully addressed by static conceptual input. Notably, performance gains plateau for larger models, indicating that further progress may require richer supervision signals, interactive fine-tuning, or stronger state-tracking mechanisms beyond static abstractions.

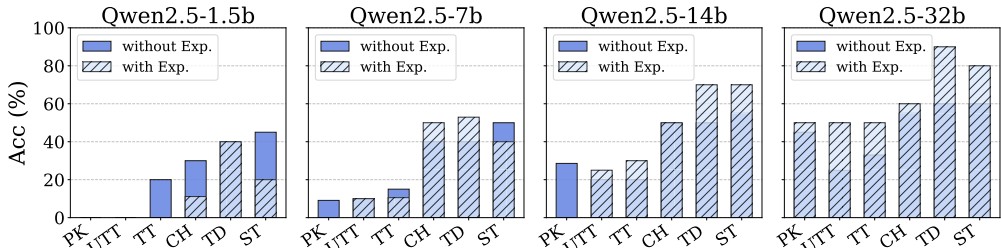

Figure 6: Win rates of Player-1 across six competitive environments from TextArena under the LfE setting.

## 6 Learning from Experience (LfE)

We investigate Learning from Experience (LfE), where models adapt by accumulating and utilizing prior interaction history. We evaluate this ability in two settings: (1) *Experience-Driven Adaptation in Competitive Games*, where agents condition on past multi-round play to adapt strategic behavior; and (2) *In-context Examples as Off-policy Exploration Supervision*, where LLMs learn from ICL-style demonstrations viewed as episodic traces.

### 6.1 Experience-Driven Adaptation in Competitive Games

**Experiment Setup.**   We adopt the same experimental setup as in Section 5.1, using TextArena [28], a suite of competitive environments featuring symbolic rules and multi-turn dynamics. The results are shown in Figure 6. Here, "with experience" refers to a setting where, during the $k$-th game, the player selects three prior game experiences from rounds 0 to $k-1$ (or all available ones if fewer than three exist), which are then used as ICL examples, allowing the model to learn from past experiences. Implementation details are provided in Appendix A.5.

**Main Result.**   Figure 6 presents win rates of Player-1 across six competitive environments in the TextArena benchmark, under two supervision regimes: without experience and with experience. Our analysis yields the following key findings: First, we observe a clear scaling trend in the efficacy of experience-based learning. For smaller models such as Qwen2.5-1.5B, the inclusion of past game trajectories generally leads to performance degradation across all environments. For example, in Checkers and Stratego, win rates drop from 0.30 to 0.11 and from 0.45 to 0.20, respectively. This suggests that low-capacity models struggle to extract relevant patterns from prior games and may treat such input as noise or distractors, leading to suboptimal decisions. Second, as model size increases, the ability to benefit from experience becomes more pronounced. Qwen2.5-7B exhibits mixed but improving trends, while Qwen2.5-14B and Qwen2.5-32B demonstrate consistent performance gains across most games when experience is incorporated. Notably, Qwen2.5-32B shows substantial improvements in complex environments like Stratego (0.6 to 0.8) and TruthAndDeception (0.6 to 0.9), indicating that larger models can better internalize structured histories and use them to inform strategy. Third, independent of experience usage, we observe a monotonic increase in win rates with model scale. This finding is consistent with results from learning from concept in Section 5.1, and confirms that larger models possess stronger general gameplay competence. Lastly, compared to concept-based generalization (Section 5.2), experience-based learning appears to be more cognitively demanding. Unlike abstract rules, game histories are long, structurally rich, and often noisy, posing challenges for information extraction and reuse. Even for Qwen2.5-32B, environments like UltimateTicTacToe show only modest gains, highlighting the inherent difficulty of experience-based generalization. These results demonstrate that (i) learning from experience is a scale-emergent ability, and (ii) its effectiveness is tightly coupled with a model's capacity to integrate, abstract, and act upon past interactions.

### 6.2 In-context Examples as Off-policy Exploration Supervisions

**Experiment Setup.**   To investigate how accumulated experience influences model behavior, we treat in-context examples as trajectories encoding implicit supervision. This setup parallels off-policy learning: unlike competitive gameplay in Section 6.1, where models adapt through active interaction

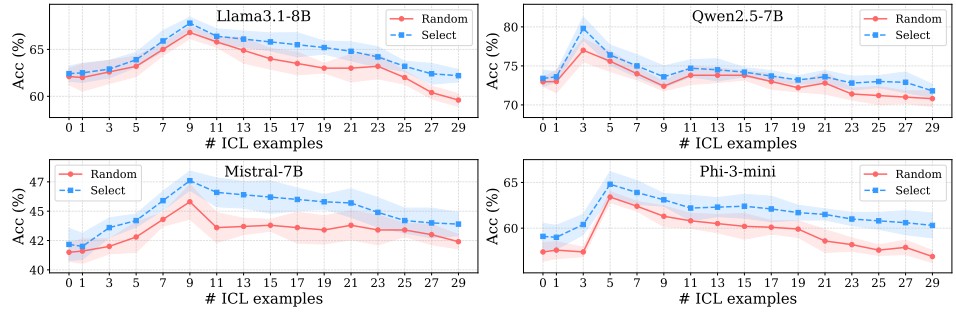

(a) Performance with different numbers of ICL examples.

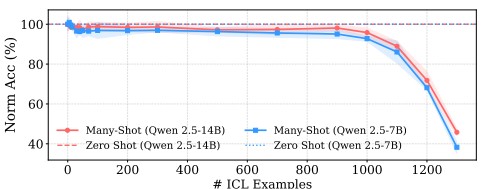

(b) Evaluating model behavior in the many-shot in-context learning setting.

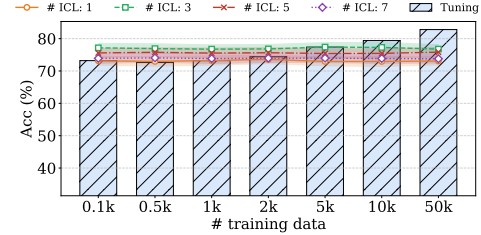

(c) Comparing in-context learning with instruction tuning under different data scales.

Figure 7: Experiments on learning from experience under off-policy exploration supervision.

(on-policy), here they condition passively on externally provided experience examples. In Figure 7a, we evaluate four base models, LLaMA3.1-8B, Qwen2.5-7B, Mistral-7B, and Phi-3-mini, under varying numbers of in-context exemplars. In Figure 7b, we scale the amount of examples up to 1300 and track the performance of Qwen2.5-14B-Instruct-1M as well as Qwen2.5-7B-Instruct-1M variants. "Norm acc" in Figure 7b denotes accuracy normalized by the model's zero-shot performance. Finally, in Figure 7c, we compare two supervision regimes: in-context learning and instruction tuning. We use MagpieMath [86], as introduced in Section 4, as the source of experience (i.e., the ICL examples), and the test set is also drawn from it. All ICL-example experiments were run 20 times, and we report the mean and standard deviation. Implementation details are provided in Appendix A.5.

**Main Result.** As shown in Figure 7a, all models exhibit a non-monotonic trend as the number of in-context examples increases: performance improves initially, peaks, and then declines. This highlights that current LLMs are effective few-shot learners but struggle to scale with longer trajectories, raising doubts about their many-shot capabilities. Figure 7b further confirms this limitation: when the number of in-context examples exceeds  900, performance drops sharply, even in larger models. This suggests that excessive examples may exceed attention capacity or interfere with generalization. Additionally, the 14B model consistently outperforms the 7B model, suggesting greater robustness to many-shot degradation. Figure 7c compares few-shot learning and instruction tuning. When data is limited, few-shot learning is highly efficient, using just 3 examples matches the performance of tuning with 5k instances. However, as data increases, instruction tuning continues to improve, while few-shot performance declines. This suggests a practical strategy: use few-shot learning in low-data regimes, and switch to tuning as more data becomes available.

## 7 LearnArena: A Benchmark Suite for General Learning Ability

**Experiment Setup.** We construct a benchmark suite that evaluates LLMs' general learning ability across three learning dimensions. The benchmark is built upon a modified version of the TextArena framework [28], where each environment is cast as a two-player game between Player-0 and Player-1. We fix Player-0 as Qwen2.5-32B and designate the evaluated LLM as Player-1. At each round $k$, Player-1 selects three experiences from the previous $k - 1$ games and submits them to Player-0 for feedback. Player-0 provides suggestions for improvement, which Player-1 incorporates in the current round, **representing Learning from Instructor**. Separately, a concise summary of the game

rules is generated by Qwen2.5-32B and given to Player-1, **capturing Learning from Concept**. In addition, Player-1 is encouraged to perform its own analysis of the selected past experiences and apply its conclusions in the current game, **reflecting Learning from Experience**. We evaluate each model-environment pair across 20 independent matches and report the average win rate of Player-1. The environments span a diverse set of strategic and social reasoning tasks, including Checkers (CH), Stratego (ST), Tic Tac Toe (TT), Truth and Deception (TD), SpellingBee (SB), SpiteAndMalice (SM), Tak (TK), and WordChains (WC). We evaluate the following models: Llama-3.1-8B, Mistral-7B-v0.3, Mistral-8B-2410, Qwen2.5-7B, Qwen2.5-14B-Instruct, Qwen2.5-32B-Instruct, Qwen3-8B, Qwen3-14B, GPT-4-o, and GPT-4-o-mini. Further implementation details are provided in Appendix A.6.

**Main Result.** As shown in Table 2, the performance of LLMs varies widely across tasks, model families, and model scales, revealing several important trends. First, GPT-4o sets a new state-of-the-art, achieving an average win rate of 0.77 across the eight tasks, significantly outperforming all other models. Notably, GPT-4o demonstrates strong and consistent results across both symbolic reasoning tasks (e.g., 0.80 on Checkers and 0.81 on Tic Tac Toe) and social reasoning environments (e.g., 1.0 on Truth and Deception),

Table 2: Win rates of Player-1 (evaluated model) across eight environments in the LearnArena benchmark. **Bold** indicates the best result for each task.

| Model | CH | ST | TT | TD | SB | SM | TK | WC | Avg. |
|---|---|---|---|---|---|---|---|---|---|
| **GPT4-o-mini** | 0.20 | 0.52 | 0.20 | **1.00** | **0.70** | 0.50 | 0.33 | 0.70 | 0.52 |
| **GPT4-o** | **0.80** | 0.61 | **0.81** | **1.00** | 0.60 | **0.82** | 0.71 | **0.80** | **0.77** |
| **Llama-3.1-8b** | 0.35 | 0.70 | 0.10 | 0.75 | 0.33 | 0.60 | 0.63 | 0.17 | 0.45 |
| **Mistral-7b** | 0.31 | 0.79 | 0.10 | 0.55 | 0.50 | 0.60 | **1.00** | 0.00 | 0.48 |
| **Mistral-8b** | 0.12 | 0.60 | 0.00 | 0.54 | 0.10 | 0.29 | 0.80 | 0.29 | 0.34 |
| **Qwen2.5-7b** | 0.39 | 0.61 | 0.08 | 0.93 | 0.20 | 0.11 | 0.10 | 0.30 | 0.34 |
| **Qwen2.5-14b** | 0.42 | 0.67 | 0.13 | 0.73 | 0.50 | 0.33 | 0.50 | 0.50 | 0.47 |
| **Qwen2.5-32b** | 0.40 | 0.72 | 0.09 | 0.83 | 0.30 | 0.50 | 0.67 | 0.60 | 0.51 |
| **Qwen3-8b** | 0.50 | **0.88** | 0.33 | 0.89 | 0.20 | 0.38 | 0.63 | 0.11 | 0.49 |
| **Qwen3-14b** | 0.50 | **0.88** | 0.50 | **1.00** | 0.50 | 0.20 | **1.00** | 0.25 | 0.60 |

showcasing its robust integration of instructional guidance, conceptual abstraction, and experiential adaptation. Its smaller variant, GPT-4o-mini, also performs competitively (0.52), matching Qwen2.5-32B and surpassing several larger models. Second, within the Qwen2.5 series, we observe scale-driven improvement: performance increases steadily from 7B (0.34) to 14B (0.47) and 32B (0.51), confirming that learning ability benefits from increased capacity, but only up to a point. The gains diminish with scale. Third, the newer Qwen3 models exhibit a significant leap in performance: Qwen3-8B reaches 0.49, while Qwen3-14B achieves 0.60, outperforming all other open-source models and even exceeding GPT-4o-mini. This indicates that beyond scale, architectural and training advancements play a crucial role in improving learning ability.

# 8    Conclusion

To address the lack of analyzing of learning ability in large language models, we propose a cognitively grounded framework that decomposes general learning into three key dimensions: learning from instructor, learning from concept, and learning from experience. This decomposition enables a principled analysis of how LLMs acquire, internalize, and apply new knowledge. Our experiments reveal that interaction enhances instruction-based learning, conceptual understanding scales with model size, and LLMs struggle in many-shot scenarios due to long-context limitations. Motivated by these findings, we introduce LearnArena, a unified benchmark designed to evaluate general learning abilities across cognitively meaningful settings. While our current evaluation focuses on domains such as mathematics and games, future work should expand the scope of LearnArena to encompass a broader range of learning contexts and conceptual domains.

# 9    Acknowledgement

This work was supported in part by the National Key R&D Program of China (Grant No.2023YFF0725001),in part by the National Natural Science Foundation of China (Grant No.92370204),in part by the guangdong Basic and Applied Basic Research Foundation (Grant No.2023B1515120057),in part by the Education Bureau of Guangzhou

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

# A  Experimental Setup

## A.1  Implementation Details

All experiments were conducted on a high-performance computing server equipped with an AMD EPYC 7V12 64-Core Processor, 512 GB of system memory, and 96 logical CPUs. The system also includes eight NVIDIA A100 GPUs, each with 80 GB of memory. Our training script is adapted from the example provided by LlamaFactory [100]. [3] Full configuration details are provided in Appendix I. The references to Llama3.1-8B, Qwen2.5-7B, Mistral-7B, Phi-3-mini, Qwen2.5-1.5B, Qwen2.5-14B, Qwen2.5-32B, Qwen2.5-72B, Mistral-8B, Qwen3-8B, and Qwen3-14B in the main text refer to the specific models: Llama-3.1-8B-Instruct, Qwen2.5-7B-Instruct, Mistral-7B-Instruct-v0.3, Phi-3-mini-128k-instruct, Qwen2.5-1.5B-Instruct, Qwen2.5-14B-Instruct, Qwen2.5-32B-Instruct, Qwen2.5-72B-Instruct, Mistral-8B-Instruct-2410, Qwen3-8B, and Qwen3-14B, respectively. Each environment in LearnArena contains 50 unique tasks and is evaluated with 20 independent runs per task under a fixed protocol. Average input lengths (tokens) are: WordChains 131, SpellingBee 63.9, SpiteAndMalice 593, Tak 739, TicTacToe 166, Stratego 839, Checkers 289, and TruthAndDeception 122.9.

## A.2  Evaluation Datasets

**SVAMP**    [63] SVAMP tasks are designed to provide a robust benchmark for evaluating the mathematical reasoning capabilities of language models on Math Word Problems (MWPs). Each task presents a short natural language narrative that describes a real-world scenario and poses a quantitative question. Previous benchmarks, such as ASDiv-A and MAWPS, were shown to permit high model performance even when critical question components were removed or word-order was disregarded, revealing the presence of shallow heuristics in existing approaches. SVAMP addresses these deficiencies by introducing examples that require models to demonstrate sensitivity to the question, robust reasoning, and invariance to structural alterations.

**MATH**    [31] MATH tasks are constructed from a dataset of 12,500 challenging competition-level mathematics problems, specifically designed to evaluate the mathematical problem-solving capabilities of language models. Each problem is accompanied by a detailed, step-by-step solution, enabling both the assessment of answer derivations and the development of explanation generation. In addition to the main benchmark, a large auxiliary pretraining dataset is provided to help teach models the fundamentals of mathematics. Despite advancements in large-scale Transformer models, accuracy on the MATH benchmark remains relatively low, highlighting the limitations of current approaches and the need for new algorithmic innovations in mathematical reasoning.

**NumGLUE**    [57] NumGLUE tasks are derived from a comprehensive multi-task benchmark designed to evaluate the numerical reasoning abilities of AI systems across eight distinct task types, each requiring some form of arithmetic understanding. The benchmark includes tasks involving commonsense and domain-specific arithmetic reasoning, quantitative comparison, fill-in-the-blank questions, reading comprehension with explicit and implicit numerical reasoning, quantitative natural language inference, and arithmetic word problems. By assessing model performance across these diverse task formats, NumGLUE provides a rigorous evaluation of both simple and complex numerical reasoning skills without leveraging explicit task-type labels during training.

**SimulEq**    [43] SimulEq tasks are based on a dataset comprising 514 test samples centered on equation solving, with a particular focus on algebraic manipulation and logical reasoning. The dataset is designed to evaluate models' proficiency in formulating and solving algebraic equations through the application of structured reasoning processes.

**AQuA**    [48] AQuA tasks are derived from the AQUA-RAT dataset, which comprises approximately 100,000 algebraic word problems accompanied by natural language rationales. Each problem includes a natural language question, five answer options (A–E) with a single correct choice, and a detailed rationale describing the solution process. This dataset enables evaluation of both answer selection and reasoning transparency in mathematical problem solving.

---

[3]`https://github.com/hiyouga/LLaMA-Factory/blob/main/examples/train_full/llama3_full_sft_ds3.yaml`

**SAT** [101] SAT tasks are sourced from the AGIEval benchmark, which is designed to evaluate the general abilities of foundation models using authentic, human-centric standardized exams. The SAT component consists of mathematics and verbal reasoning problems that mirror those found in real-world college entrance assessments. By incorporating these tasks, AGIEval provides a robust framework for assessing models' understanding, reasoning, and calculation skills at a human-competitive level. Recent results show that state-of-the-art models, such as GPT-4, can achieve or surpass average human performance on SAT tasks, highlighting both the progress and current limitations of large language models in solving complex, human-oriented problems.

**MMLU-Math** [30] MMLU-Math tasks are part of a comprehensive multitask benchmark designed to evaluate a model's accuracy across a wide range of academic subjects, with a specific focus on elementary and advanced mathematics. The benchmark requires models to demonstrate both broad world knowledge and sophisticated problem-solving skills. While recent large language models show improvements over random-chance accuracy, substantial gaps remain in achieving expert-level performance, particularly in mathematics. MMLU-Math enables detailed analysis of a model's mathematical proficiency and helps to identify critical areas requiring further advancement.

**MagpieMath** [86] MagpieMath is a large-scale dataset generated via the Magpie pipeline, which synthesizes high-quality mathematical instruction data through auto-regressive sampling from Qwen2.5-Math. Unlike prior instruction tuning datasets that depend on seed prompts or handcrafted templates, Magpie employs a pre-query-based sampling approach to elicit diverse and pedagogically rich instruction data directly from aligned LLMs. The resulting dataset spans a wide range of mathematical skills, including arithmetic, algebra, and multi-step reasoning, and serves as a robust resource for evaluating instruction-based learning in mathematical domains.

**Poker** [28] is a strategic two-player Texas Hold'em game where players compete by betting chips based on their hole cards and community cards. Each player aims to win chips by either having the best hand at showdown or making their opponent fold. The game follows standard Texas Hold'em rules with a fixed number of rounds and betting structures.

**Ultimate Tic Tac Toe** [28] is a strategic two-player game that combines the classic Tic Tac Toe with an added layer of complexity. Players aim to win three micro boards in a row (horizontally, vertically, or diagonally) on the macro board, which tracks the outcomes of individual micro boards. Each move influences the opponent's next playable micro board, creating dynamic and strategic gameplay. This environment implements the full rules of Ultimate Tic Tac Toe, including valid move enforcement, micro board and macro board win detection, and a clear rendering of the board state for agent-based gameplay and experimentation.

**Checkers** [28] (also known as Draughts) is a turn-based game played on an 8×8 board. Each player controls a set of pieces placed on alternating dark squares. Pieces move diagonally forward and can capture opponent pieces by jumping over them. A piece that reaches the opposite side of the board becomes a "King," gaining the ability to move backward as well. The game ends when one player's pieces are all captured or when a player has no valid moves left.

**Stratego** [28] is a two-player strategy board game where players aim to capture their opponent's Flag or eliminate all of their movable pieces. Each player strategically places their army, consisting of pieces of varying ranks, on a 10x10 grid, with special rules for movement and battle. Bombs act as immovable traps, while Scouts can move multiple spaces in a straight line. The game involves hidden information, where a player's pieces are unknown to their opponent until revealed in battle. This environment simulates the full rules of Stratego, including strategic placement, battle resolution, and dynamic board rendering, providing a robust setup for agent-based gameplay and experimentation.

**TicTacToe** [28] (also known as Noughts and Crosses) is a classic two-player game played on a 3×3 grid. Players take turns marking cells with their symbol ('X' or 'O'), with the objective of placing three of their marks in a horizontal, vertical, or diagonal row. This implementation provides a complete game environment with move validation, win detection, and draw conditions.

**Truth and Deception** [28] is a two-player social deduction game where one player (the Deceiver) attempts to mislead the other player (the Guesser) about which of two facts is correct. The players

engage in a conversation of limited turns, during which the Deceiver strategically shares information to encourage the Guesser to choose the incorrect fact. At the end of the conversation, the Guesser must determine which fact is true. The game tests persuasion, critical thinking, fact verification, and strategic communication in a structured environment with clear objectives for both roles.

**Spelling Bee** [28] is a two-player word game where players take turns creating valid English words using only a limited set of allowed letters. The challenge increases as the game progresses because each new word must be at least as long as the previous word. Players must balance vocabulary knowledge with strategic thinking to outmaneuver their opponent by forcing them into increasingly difficult positions. This implementation features frequency-weighted letter selection to ensure balanced and playable letter sets.

**Spite and Malice** [28] is a two-player competitive card game that combines elements of solitaire and strategic play. Each player has their own payoff pile that they aim to deplete first to win. Players can play cards to shared center piles in ascending sequence (Ace to Queen), with Kings serving as wild cards. The game involves careful resource management, opportunistic card placement, and strategic blocking to prevent your opponent from emptying their payoff pile. This implementation features a complete deck management system, discard piles, and a hand limit of five cards.

**Tak** [28] is a two-player abstract strategy board game. Players aim to build a continuous road connecting two opposite edges of the board while blocking their opponent's attempts. The game involves placing and moving pieces of various types—Flat Stones, Standing Stones (Walls), and Capstones—each with unique abilities. This environment simulates the full rules of Tak, including flexible board sizes, road-building mechanics, and dynamic board rendering, offering a rich platform for agent-based gameplay and experimentation.

**Word Chains** [28] is a turn-based word game where two players take turns providing valid English words that start with the last letter of the previous word. The game enforces rules such as no word repetition and requires valid English words. Players lose if they fail to provide a valid word.

**LogicGrid** [52] LogicGrid tasks are constructed as logic puzzles that require participants to perform grid-based deduction and symbolic reasoning. These tasks are specifically designed to evaluate the systematic application of abstract rules within constrained environments. All LogicGrid tasks presented in this work are newly developed adaptations based on established principles from classic puzzle collections.

**NaLogic** [23] NaLogic tasks are narrative-driven logic puzzles that require solvers to extract and apply logical constraints embedded within rich, contextualized storytelling. These tasks are directly inspired by literary puzzle formats, and they are specifically designed to assess the integration of logical reasoning within narrative comprehension. All NaLogic tasks included in this work are systematically derived from established approaches found in classic puzzle literature.

**Plan** [79] Plan tasks are adapted from classical scenarios formulated within the Planning Domain Definition Language (PDDL), including representative domains such as Gripper, Barman, Blocksworld, and Tyreworld. In these tasks, symbolic states are systematically translated into natural language descriptions to enable seamless interaction with large language models. Progress is quantitatively tracked by computing state-goal matching scores, thus providing a robust measure of task completion and logical reasoning within dynamic planning environments.

**AlfWorld** [74] AlfWorld tasks require agents to perform everyday household activities that necessitate both exploration and commonsense reasoning. AlfWorld leverages interactive TextWorld environments that are closely aligned with the embodied scenarios found in the ALFRED dataset. This alignment enables agents to develop and reason about high-level policies within an abstract, text-based space, prior to executing embodied tasks that demand low-level actuation in physical or simulated environments.

**ScienceWorld** [82] ScienceWorld tasks require agents to comprehend and execute a variety of scientific procedures, including activities such as measuring physical properties and manipulating

laboratory equipment. Each task is accompanied by carefully reannotated subgoals designed to more accurately capture incremental progress, thereby enabling a granular evaluation of scientific reasoning and procedural understanding.

**BabyAI**    [10] BabyAI tasks require agents to navigate and interact within grid-world environments, leveraging newly introduced textual action representations and re-annotated subgoals. These modifications are designed to facilitate more nuanced agent-environment interaction and provide finer-grained evaluation of task completion in language-conditioned navigation settings.

**AlpacaEval**    [22] AlpacaEval tasks consist of a diverse set of 805 instructions, systematically aggregated from multiple benchmark sources. Specifically, the collection includes 252 instructions from the self-instruct test set[83], 188 from the Open Assistant test set (OASST), 129 from the Anthropic Helpful Test Set [102], 80 from the Vicuna test set [12], and 156 from the Koala test set [80]. This comprehensive assembly is designed to support the broad evaluation of the instruction-following capabilities across a wide range of language models.

**Vicuna**    [12] Vicuna tasks consist of 80 test instructions, systematically organized into eight distinct categories: Fermi problems, common sense reasoning, role-play scenarios, coding, mathematics, writing tasks, counterfactuals, knowledge assessment, and generic questions. This categorical organization is designed to facilitate a thorough evaluation of multiple dimensions of chatbot performance. Prior research suggests that the Vicuna dataset predominantly features instructions of comparatively lower difficulty and complexity[85]. In this study, the Vicuna test set is used to specifically assess the performance of large language models across these various types of instruction.

**Self-Instruct**    [83] Self-Instruct tasks consist of 252 human-authored test instructions, each paired with a carefully crafted reference output. This test set is meticulously curated to represent the real-world applicability of instruction follower models, encompassing a wide range of domains such as email composition, social media engagement, productivity software, and coding tasks. The instructions exhibit substantial diversity in both style and format, including various task lengths and a spectrum of input/output types such as bullet points, tables, code fragments, and mathematical equations. In this study, the Self-Instruct test set is utilized to rigorously evaluate the model's proficiency in adhering to precise instructions across heterogeneous domains.

**Wizardlm**    [85] Wizardlm tasks are derived from a large-scale instruction dataset initially seeded with 52,000 instructional examples from Alpaca, subsequently expanded through $M = 4$ evolutionary cycles to produce 250,000 unique instructions. In each cycle, for every instruction, six new prompts are generated—five via in-depth evolution and one through in-breadth evolution—of which one prompt is randomly selected. Responses are then generated using ChatGPT, culminating in $52,000 \times 4 \times 3 = 624,000$ instruction-response pairs. The training subset utilized for the Evol-Instruct dataset comprises 70,000 of these instructions. The test set, consisting of 218 instructions, is curated from diverse sources, including open-source projects and online forums, and encompasses 29 distinct skill areas identified from real-world human tasks.

**ARC**    [13] ARC tasks are derived from a dataset comprising 7,787 English-language science exam questions sourced from a variety of educational materials, including licensed content provided by a research partner affiliated with AI2. Each question is presented in a multiple-choice format, typically with four answer options, and spans a range of grade levels. The dataset is partitioned into a Challenge Set containing 2,590 "hard" questions—defined as those that cannot be correctly answered by either retrieval-based or co-occurrence-based baseline methods—and an Easy Set containing 5,197 questions of comparatively lower difficulty. This structure supports comprehensive evaluation of reasoning and scientific understanding across different levels of complexity.

**HellaSwag**    [93] HellaSwag tasks are derived from a challenging benchmark for commonsense natural language inference, wherein models must select the most plausible continuation for a given event description. For example, given an event such as "A woman sits at a piano," the model must choose the most likely follow-up, such as "She sets her fingers on the keys." Despite the introduction of advanced pretrained models like BERT, which achieved near human-level performance on previous tasks, HellaSwag demonstrates that even state-of-the-art models still struggle with robust

commonsense reasoning. The dataset is constructed using Adversarial Filtering, a methodology where a series of discriminators iteratively select adversarially challenging, machine-generated distractors. While these questions are trivial for humans (accuracy >95%), state-of-the-art models frequently misclassify them (accuracy <48%).

**WinoGrande** [67] WinoGrande tasks are derived from a dataset comprising 44,000 problems, inspired by the Winograd Schema Challenge [41], but specifically adjusted to enhance both scale and robustness against dataset-specific biases. Each task is presented as a fill-in-the-blank question with binary options, requiring the participant to select the correct answer based on contextual commonsense reasoning.

**MMLU** [30] MMLU tasks are derived from the benchmark introduced by Hendrycks et al., evaluating a model's ability to answer multiple-choice questions across a broad range of academic subjects. In the original format, each question presents the answer choices within the context, and the model is required to produce the answer letter (e.g., A, B, C, D) as the continuation. The *mmlu-continuation* variant adopts a cloze-style format, omitting the answer choices in the context and requiring the full answer choice in the continuation. The *mmlu-generation* variant is similar to the original, but the model is tasked with generating the correct answer letter directly. These variants collectively enable a nuanced evaluation of language models' reasoning and generation capabilities across diverse knowledge domains.

**GSM8k** [16] GSM8k tasks are drawn from a dataset of 8,500 high-quality, linguistically diverse grade school math word problems, designed to evaluate multi-step mathematical reasoning in language models. Despite the conceptual simplicity of the problem distribution, state-of-the-art transformer-based models consistently struggle to achieve high performance on this benchmark. The GSM8k dataset thus provides a rigorous diagnostic tool for analyzing the limitations of current models and supporting further research in mathematical reasoning.

**Intlaw** [61, 62] Intlaw tasks are derived from L. Oppenheim's two-volume *International Law: A Treatise*, and are designed to assess understanding of both foundational and applied principles in international law. Volume 1 emphasizes legal theory, sources of law, and the concept of state sovereignty, while Volume 2 explores topics such as conflict, neutrality, and mechanisms for peaceful dispute resolution. This dataset enables evaluation of models' capabilities in legal reasoning, state interactions, and comprehension of diplomatic and legal frameworks.

**Willidea** [69, 70, 71] Willidea tasks are constructed from all three volumes of Arthur Schopenhauer's *The World as Will and Idea*, focusing on the evaluation of philosophical reasoning and metaphysical analysis. The dataset challenges models to interpret complex philosophical arguments, engage with Kantian critique, and reason across abstract conceptual structures related to perception, will, and reality. Through these tasks, the ability of language models to comprehend and analyze dense philosophical discourse is rigorously assessed.

**Ancsoc** [58] Ancsoc tasks are derived from Lewis H. Morgan's *Ancient Society* and are designed to assess a model's understanding of the evolution of human civilization through the stages of savagery, barbarism, and civilization. The dataset emphasizes analysis of social structures, cultural development, and key concepts in anthropological theory. Through these tasks, models are evaluated on their capacity for historical analysis and cross-cultural reasoning.

**Poliec** [65, 66] Poliec tasks are based on Wilhelm Roscher's two-volume *Principles of Political Economy* and focus on the historical development of economic systems, classification of income, and the influence of human behavior on economic theory. The dataset is designed to assess models' abilities to reason through complex economic frameworks, drawing on both historical context and theoretical foundations.

### A.3 Details of Learning from the Instructor

We evaluate instruction-based learning ability across five distinct domains: Mathematics, Law, Philosophy, Anthropology, and Economy. Each domain is selected to reflect different facets of

instruction-following behavior, ranging from structured problem-solving to conceptual understanding. During training, we additionally use the MAGPIEMATH dataset [86], which provides large-scale, high-quality mathematical instruction data synthesized from Qwen2.5-Math. It is a subset of the dataset available at this link [4]. The prompt can be found in Appendix L. We used eight representative test benchmarks: GSM8K [16], SVAMP [63], MATH [31], NumGLUE [57], SimulEq [43], AQuA [48], SAT [101], and MMLU-Math [30]. These datasets cover a range of mathematical competencies from grade-school arithmetic to advanced problem-solving, including multi-step reasoning, equation solving, and standardized testing skills. In the Law domain, we use the Intlaw dataset [61, 62]. It covers core topics such as sovereignty, conflict, and dispute resolution, and evaluates models' legal reasoning and understanding of international legal systems. In the Philosophy domain, we use the Willidea dataset [69, 70, 71], which focuses on metaphysics, perception, and will, testing models' ability to interpret abstract arguments and philosophical reasoning. In the Anthropology domain, we use the Ancsoc dataset [58], which evaluates understanding of social evolution, cultural systems, and anthropological theory. In the Economy domain, we use the Poliec dataset [65, 66], which emphasizes historical and theoretical reasoning in value, income, and economic behavior. Across these four non-math domains, question formats included fill-in-the-blank (fill), judgment (judge), multiple-choice (multi-choice), open-ended (open), and single-choice (single-choice). We provide representative cases of each question format in Appendix J. Further dataset details are provided in Appendix A.2

## A.4    Details of Learning from the Concept

We evaluate models' ability to learn from structured conceptual information across a variety of tasks that require rule interpretation, logical reasoning, and symbolic planning. Our experiments focus on two complementary settings: multi-agent competitive environments and logic-based planning tasks. In each case, we compare model performance with and without access to high-level conceptual summaries that describe task-specific rules, goals, or strategies. These summaries are generated by a stronger teacher model and provided as auxiliary context to the learner model. We first study conceptual learning in strategic settings using the TEXTARENA framework [28], a collection of multi-agent environments characterized by symbolic rules and multi-turn dynamics. Each game involves two players, Player-0 and Player-1. We fix Player-0 as Qwen2.5-32B and vary Player-1 across four model scales: Qwen2.5-1.5B, 7B, 14B, and 32B. Prior to gameplay, Player-1 either receives no guidance (without Concept) or is given a natural language description of the game rules and strategies (with Concept) generated by Qwen2.5-32B. For each environment and model combination, we run 20 matches and report the average win rate of Player-1. The evaluation environments include CHECKERS (CH), POKER (PK), STRATEGO (ST), TIC TAC TOE (TT), TRUTH AND DECEPTION (TD), and ULTIMATE TIC TAC TOE (UTT). Performance is measured as the win rate of Player-1 against the fixed opponent. We then evaluate rule-based generalization across a set of logic and planning tasks. The LOGICGRID and NALOGIC tasks consist of structured and narrative logic puzzles [52, 23], requiring deduction under symbolic constraints. PLAN [79] adapts classical PDDL-style planning domains (e.g., Gripper, Barman) into natural language to assess symbolic action reasoning. Additionally, we include ALFWORLD [74], SCIENCEWORLD [82], and BABYAI [10] tasks, which involve procedural and goal-oriented behaviors requiring commonsense reasoning and spatial understanding. For the concept-injected setting, we use Qwen2.5-72B to generate static conceptual summaries tailored to each task. These summaries are provided as additional input to the learner model during evaluation. Further dataset details are provided in Appendix A.2

## A.5    Details of Learning from the Experience

We investigate how language models adapt their behavior based on prior experiences. Our experiments are designed to capture different aspects of experience-based learning, ranging from competitive multi-turn interactions to static in-context supervision. In all settings, models are exposed to past interaction traces or curated examples, which serve as proxies for experiential knowledge. We evaluate two complementary scenarios: (1) interactive adaptation through self-selected experience in multi-agent games, and (2) passive conditioning on externally provided in-context examples. These two settings align conceptually with classical learning paradigms: the first resembles on-policy learning, where the model actively explores and adapts through its own interactions; the second

---

[4]`https://huggingface.co/datasets/Magpie-Align/Magpie-Qwen2.5-Math-Pro-300K-v0.1`

mirrors off-policy learning, where the model passively leverages prior trajectories without influencing their generation. Together, they allow us to explore whether LLMs can extract useful patterns from previous data. We adopt the same multi-agent environment setup described in Section 5.1, using TEXTARENA [28], a suite of competitive games involving symbolic rules and multi-turn dynamics. The results are shown in Figure 6. In this setting, "with experience" refers to a scenario where, during the $k$-th game, the model selects three prior experiences from rounds 0 to $k-1$ (or all available ones if fewer than three exist). These samples are used as in-context learning (ICL) examples, allowing the model to draw on its past trajectories for real-time adaptation. Performance is measured by the average win rate of Player-1 across 20 independent matches. This setup captures whether models can benefit from self-curated experiential memory in competitive decision-making. To further analyze how accumulated experience affects model behavior, we treat in-context examples as off-policy trajectories that encode supervision signals. This setup parallels off-policy learning: unlike competitive gameplay in Section 6.1, where models adapt through active interaction (on-policy), here they condition passively on externally provided experience examples. These exemplars are not generated by the current model but are instead drawn from prior runs or curated datasets. We use MagpieMath [86], introduced in Section4, as the source of experience for in-context learning, with both the ICL examples and test set drawn from this dataset. All in-context experiments are repeated 20 times using different random seeds. We report the mean and standard deviation for each result. Further dataset details are provided in Appendix A.2

## A.6    Details of LearnArena

We design a benchmark suite to evaluate the general learning ability of large language models across three distinct learning paradigms: Learning from Instructor, Learning from Concept, and Learning from Experience. Each paradigm is operationalized within the same unified game-based evaluation setting, adapted from the TextArena framework [28]. In this setup, each environment is framed as a two-player game involving Player-0 and Player-1. Player-0 is fixed as Qwen2.5-32B and serves as the consistent teacher or opponent across all tasks. Player-1 is the evaluated model, which varies across runs. At each round $k$, Player-1 selects three prior experiences from rounds 0 to $k-1$ and submits them to Player-0 to receive feedback. Player-0 then returns suggestions for improvement. This interactive feedback process constitutes the *Learning from Instructor* component of the benchmark. In parallel, a concise natural language summary of the environment's rules is generated by Qwen2.5-32B and given to Player-1 prior to gameplay. This summary includes key mechanics, legal moves, and basic strategies for the task, enabling models to condition on explicit rule representations. This serves as the *Learning from Concept* component. Player-1 is also encouraged to analyze its own past experience samples and draw inferences to improve future performance. These self-selected experiences are used as in-context examples during gameplay, forming the *Learning from Experience* component. The prompt can be found in Appendix K. We evaluate each model-environment pair over 20 independent matches and report the average win rate of Player-1. The benchmark includes a diverse set of environments designed to test strategic reasoning, symbolic planning, and adaptive behavior. The selected games include CHECKERS (CH), STRATEGO (ST), TIC TAC TOE (TT), TRUTH AND DECEPTION (TD), SPELLINGBEE (SB), SPITEANDMALICE (SM), TAK (TK), and WORDCHAINS (WC). We evaluate the following models as Player-1: LLaMA-3.1-8B, Mistral-7B-v0.3, Mistral-8B-2410, Qwen2.5-7B, Qwen2.5-14B-Instruct, Qwen2.5-32B-Instruct, Qwen3-8B, Qwen3-14B, GPT-4-o-mini, and GPT-4-o. Each model plays all environments under the same setup, enabling controlled and comparable assessment across architectures and scales. The details of dataset are provided in Appendix A.2 Further implementation details, including prompt formats, sampling settings, and game environment configurations, are provided in the released codebase.

**Comparison with Existing Benchmarks.**    To assess the uniqueness and relevance of our benchmark, we compare it with a broad set of widely used evaluation benchmarks by computing Spearman rank correlations between system-level rankings. These benchmarks include instruction-following datasets such as AlpacaEval [22], Vicuna [12], Self-Instruct [83], and WizardLM [85], as well as standard task-specific evaluations including ARC [13] (science reasoning), HellaSwag [93] and Winogrande [67] (commonsense inference), GSM8K [16] (mathematical reasoning), and MMLU [30] (broad-domain knowledge).

As shown in Figure 8, our benchmark shows low correlation with all existing benchmarks, with the highest only reaching a moderate level (0.66), indicating it captures distinct aspects of model

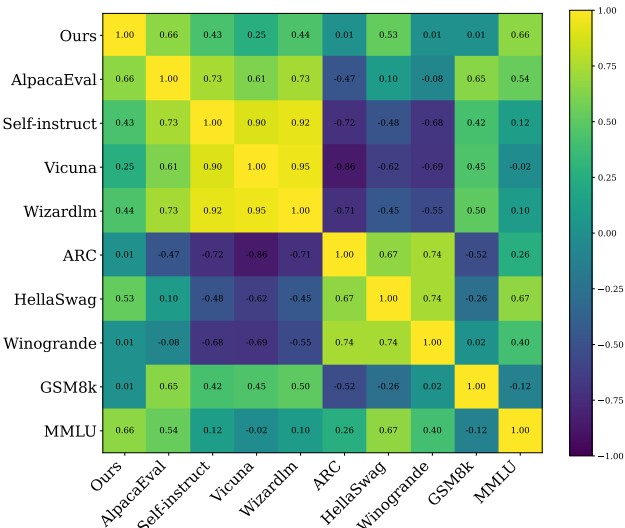

Figure 8: Spearman rank correlation matrix between our benchmark and nine existing benchmarks.

performance. Among all benchmarks, our strongest correlations are with AlpacaEval ($\rho = 0.66$) and MMLU ($\rho = 0.66$), followed by HellaSwag ($\rho = 0.53$), WizardLM ($\rho = 0.44$), and Self-Instruct ($\rho = 0.43$). The correlation with Vicuna is relatively low ($\rho = 0.25$), and correlations with ARC, Winogrande, and GSM8K are near zero. These results indicate that while there is some overlap between our benchmark and existing instruction-following or knowledge-based tasks, our evaluation framework provides a different and complementary signal. This divergence is expected, given the design objective of our benchmark. Unlike most prior benchmarks that focus on static task-solving or narrow instruction-following performance, our benchmark is explicitly constructed to evaluate general learning ability. It is grounded in a cognitive framework that integrates three key dimensions: learning from instructor, learning from concept, and learning from experience. However, our benchmark does not treat these dimensions as isolated tasks or modules. Instead, we embed them holistically across the benchmark's task suite, such that model success requires simultaneously demonstrating the ability to absorb guidance, abstract rules, and dynamic feedback. As a result, our benchmark presents a more integrated and realistic test of adaptive intelligence, resembling how learning occurs in natural human settings. The varying degrees of correlation across external benchmarks further illuminate how different datasets capture different cognitive demands. Benchmarks like ARC focus on domain-specific problem-solving (e.g., science) with fixed question-answer formats. These tasks require correctness but do not test a model's ability to update its behavior through interaction or abstraction, which likely explains the near-zero correlation with our benchmark. On the other hand, AlpacaEval and WizardLM emphasize open-ended instruction-following, which partially aligns with our learning-from-instructor dimension. MMLU, though knowledge-centric, requires broad generalization across domains and may tap into conceptual understanding and knowledge transfer, hence the higher correlation. These findings confirm that our benchmark captures a distinct and underexplored aspect of LLM evaluation by assessing general-purpose learning through the integrated use of instruction, abstraction, and adaptation.

## B  Opponent Robustness and Extended Evaluation

LearnArena in the main text fixes Player-0 to Qwen2.5-32B. To assess robustness to opponent choice and reduce the risk that models overfit to a single opponent's idiosyncrasies, we repeated the evaluation with three alternative Player-0 models—Phi-4, GPT-4o-mini, and GPT-4o. For each fixed Player-0, we measured Player-1 performance on four environments (Checkers, Stratego, SpellingBee, SpiteAndMalice) under the same protocol as the main paper, averaged win rates across the four environments to obtain a per–Player-1 score, ranked Player-1 models by this score, and computed the Spearman rank correlation between this ranking and the baseline ranking obtained with Player-0 = Qwen2.5-32B.

Table 3: Player-0 = Phi-4. Player-1 win rates on four environments.

| Player-1 | CH | ST | SB | SM |
|---|---|---|---|---|
| gpt-4o | 1.00 | 0.90 | 0.65 | 1.00 |
| gpt-4o-mini | 0.65 | 0.88 | 0.86 | 0.72 |
| llama-3.1-8b | 0.65 | 0.70 | 0.50 | 0.75 |
| mistral-8b | 0.40 | 0.75 | 0.28 | 0.69 |
| qwen2.5-7b | 0.70 | 0.75 | 0.35 | 0.35 |
| qwen3-8b | 0.75 | 0.65 | 0.50 | 0.70 |

Table 4: Player-0 = GPT-4o-mini. Player-1 win rates on four environments.

| Player-1 | CH | ST | SB | SM |
|---|---|---|---|---|
| gpt-4o | 0.80 | 1.00 | 0.65 | 1.00 |
| gpt-4o-mini | 0.50 | 0.55 | 0.45 | 0.55 |
| llama-3.1-8b | 0.25 | 0.85 | 0.10 | 0.70 |
| mistral-8b | 0.15 | 0.85 | 0.20 | 0.45 |
| qwen2.5-7b | 0.45 | 0.70 | 0.25 | 0.35 |
| qwen3-8b | 0.40 | 0.95 | 0.55 | 0.50 |

Across these alternative opponents, the rank-order agreement with the baseline is high. The Spearman correlations with the Qwen2.5-32B–based ranking are $\rho_{\text{Phi-4}} = 1.000$, $\rho_{\text{GPT-4o-mini}} = 0.829$, $\rho_{\text{GPT-4o}} = 0.714$. These values indicate that relative model rankings are largely preserved across substantially different opponents. In practice, stronger opponents compress absolute win rates yet retain the broad ordering of Player-1 models, which is useful for difficulty calibration while maintaining comparative validity. Taken together, these results suggest that the main conclusions of the benchmark are not overly sensitive to the specific choice of Player-0. While average win rate remains a clear and discriminative summary of success in dynamic environments, it is a coarse measure of learning. To provide a more granular view of learning dynamics in future versions of LearnArena (LearnArena), we will augment reporting with process-level signals that capture how models adapt, follow guidance, and reuse experience. Specifically, we will measure adaptation speed through per-round improvement and rounds-to-threshold; instruction adherence through agreement between instructor feedback and subsequent actions under automatic alignment checks; concept utilization through rule compliance and strategy-template coverage when conceptual summaries are provided; experience reuse through the recurrence of salient patterns from selected prior trajectories in current decisions; and stability and robustness through variance across seeds and sensitivity to controlled noise injected into instructions, concepts, and experiences.

## C  Many-Shot Behavior and Long-Context Models

This section examines whether degradation in many-shot settings is merely a consequence of poor long-context capability. In the many-shot experiments reported in Section 6.2, we evaluate Qwen2.5-7B/14B-Instruct-1M, which support up to 1M-token windows and are explicitly designed for long-context usage. Across these models, performance follows a clear non-monotonic trend: accuracy improves as in-context examples increase, peaks at a small number of shots (3–9 depending on model size), and then declines as the prompt grows longer. This indicates that, even with extended context windows, current models remain effective few-shot learners while many-shot generalization remains challenging. To further validate this pattern with a frontier model under tractable cost, we additionally evaluate GPT-4o-mini (128k context) following the protocol in Section 6.2 and the setup of [4], using 4-shot as a reference and scaling up to 100 examples. On MagpieMath, we observe 71.2 (0-shot), 71.4 (4-shot), 72.8 (10-shot), a peak at 75.2 (15-shot), followed by 71.2 (20-shot), 70.8 (30-shot), 70.0 (50-shot), 70.6 (80-shot), and 70.6 (100-shot). The rise-then-fall trajectory replicates the central empirical regularity documented in prior work. The observations above are consistent with [4]. In their Section 3.1, Figure 7 (left), Gemini 1.5 Pro on MATH-500 with ground-truth exemplars exhibits the same shape: performance rises as the number of shots increases, peaks at roughly 125 shots, and then declines as the context grows further. Although [4] truncates at 500 shots, the downward trend is already evident. In our experiments on MagpieMath, when we extend beyond 30–100 examples

the decline persists and eventually dips below the 4-shot baseline, noting that the absolute scores differ across datasets and evaluation protocols. Taken together, evaluations spanning Qwen2.5-7B, Qwen2.5-14B, GPT-4o-mini, and previously reported results for Gemini 1.5 Pro all exhibit the same qualitative regularity: performance improves, peaks, and then degrades as the number of in-context examples grows. As model capability increases, the peak tends to occur at a larger shot count and the subsequent degradation is slower, suggesting that stronger long-context handling delays but does not eliminate many-shot brittleness. These results reinforce that long-context capacity is necessary but not sufficient; robust many-shot generalization remains an open challenge.

## D Clarifying "Passive Consumption" (LfI) vs. "Experience-Driven Adaptation" (LfE)

Although both settings can appear as few-shot style prompts, they differ in intent and supervision: LfI (passive consumption) supplies instructional content—worked solutions, step-by-step rationales, or corrections—from a strong instructor, providing pedagogically curated guidance; formally, the context is $\mathcal{C}_t = \mathcal{I}_t$ and prediction follows $f_\theta(x, \mathcal{I}_t) \to \hat{y}$. LfE (experience-driven adaptation) instead presents episodic traces—raw input–output pairs, partial trajectories, or outcomes—without didactic restructuring, requiring the model to infer what to abstract or discard; here $\mathcal{C}_t = \tau_t = \{s_1, \ldots, s_k\}$ and prediction follows $f_\theta(x, \tau_t) \to \hat{y}$. In our experiments, LfI uses instructor-produced solutions/rationales, whereas LfE uses selected histories from earlier rounds as off-policy trajectories, reflecting guided transfer versus autonomous abstraction from unstructured experience.

## E Rationale for Dataset Choices Across Learning Dimensions

The evaluation follows a two-stage design. Sections 4–6 isolate and validate each learning mechanism under task families that provide the right supervision affordances and minimal cross-factor interference. Section 7 then integrates all three mechanisms in a unified setting via LearnArena, enabling a holistic assessment. A single fixed dataset for all experiments would entangle constructs and reduce interpretability, whereas the staged design improves construct validity by matching task structure to the learning signal being tested. For Learning from Instructor (Section 4), mathematics is used because it naturally supports a tutor–learner paradigm with high-fidelity supervision. Problems admit worked solutions and stepwise rationales, enabling clear, pedagogically oriented instructional signals from a strong instructor (e.g., Qwen2.5-Math-72B) and objective evaluation of whether instruction improves the learner. For Learning from Concept (Section 5), domains are chosen where structured knowledge can be injected and audited. Competitive environments (TextArena) expose explicit rules and strategy templates; logic puzzles (LogicGrid, NaLogic) emphasize formal constraints; and planning/agent tasks (Plan, AlfWorld, ScienceWorld, BabyAI) probe procedural abstractions and generalization. These settings make conceptual cues precise, testable, and meaningfully tied to downstream behavior. For Learning from Experience (Section 6), the focus is on adaptation from trajectories rather than instruction. Two complementary regimes are used: on-policy experience in competitive games, where histories accumulate across rounds and must be exploited under stochastic interaction, and off-policy experience via in-context examples, where pre-collected traces serve as episodic supervision without didactic restructuring. This separation targets the specific cognitive demands of experience integration. Section 7 introduces LearnArena as a unified benchmark that combines instructional feedback, conceptual summaries, and experience selection within shared game environments. The unified evaluation complements the isolated studies by testing whether the three learning mechanisms compose in realistic multi-agent settings, while the earlier sections justify that each mechanism is measurable and interpretable in its most suitable domain.

## F Why Three Perspectives? Scope and Relationship to Self-Critique

The tripartite decomposition—Learning from Instructor (LfI), Learning from Concept (LfC), and Learning from Experience (LfE)—is chosen as a cognitively grounded and minimally overlapping

basis that captures the principal supervision channels through which learning occurs. LfI models didactic guidance delivered by an external agent via demonstrations, explanations, or corrections (instructional intent; [19, 73]); LfC captures the use of declarative, abstract structure (rules, definitions, schemata) to enable transfer and constraint-based generalization ([25, 5]); LfE characterizes adaptation from accumulated trajectories and outcomes under interaction or replay, without curated pedagogy ([42]). These axes align with long-standing accounts in cognitive psychology and education that separate guided instruction, conceptual abstraction, and experiential learning as complementary pathways ([15, 42, 25]). Self-critique and related practices (reflection, self-rewarding, self-refinement) fit naturally within this framework rather than constituting a separate axis. When a model critiques, edits, or rewards its own outputs, the instructional channel is internalized: the model acts as its own instructor while still supplying pedagogically structured signals, which places self-critique as an instance of LfI with the instructor collapsed into the learner. In settings where the model instead mines unstructured past attempts or rollouts to improve future behavior, the mechanism corresponds to LfE. Thus, the proposed three perspectives provide a coherent basis that cleanly factorizes the supervision source (instructor vs. concept vs. experience) while accommodating practical variants—external or internal—within the same taxonomy.

## G  Limitations

While our work provides a structured perspective on the learning capabilities of large language models, we acknowledge that our evaluations are conducted in synthetic and benchmark-driven environments. As such, the implications of improved learning abilities in real-world settings, such as education, healthcare, or human-AI collaboration, are not directly explored. Moreover, while better learning may enhance LLM adaptability, it may also raise concerns around user over-reliance, unintended memorization, or alignment with user intent in high-stakes applications. Our framework is intended as a diagnostic tool for understanding LLM behavior, not as a comprehensive measure of safety, fairness, or trustworthiness. We hope future work will extend our approach to include evaluations grounded in real-world impact, especially in socially sensitive domains.

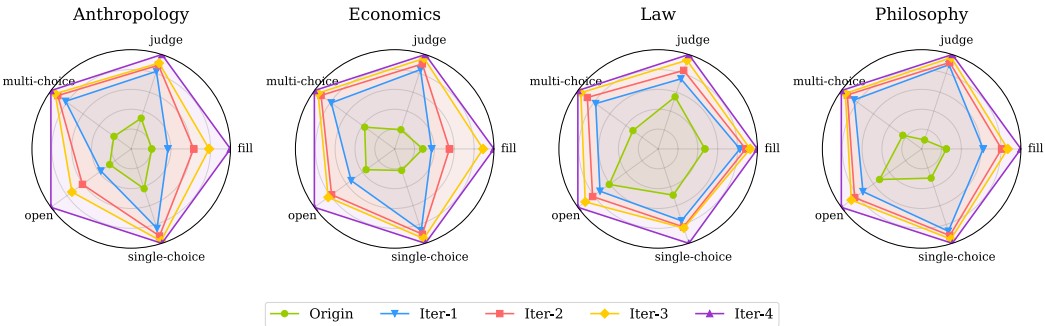

Figure 10: Model performance across five question types (fill-in-the-blank, judgment, multiple-choice, open-ended, single-choice) over four iterations. Performance consistently improves as training data becomes cleaner through selection.

## H  Learning with Useful Knowledge Selection

**Experiment Setup.**  Humans naturally develop the ability to select high-quality learning materials based on prior experience. In this section, we investigate whether LLMs possess a similar capacity to identify and prefer informative over noisy data during training. We evaluate this ability across four non-mathematical domains, Anthropology, Economics, Law, and Philosophy, where domain knowledge is derived from canonical classical texts, including works on international law [61, 62], philosophy [69, 70, 71], anthropology [58], and economics [65, 66]. Each domain includes five question types: fill-in-the-blank (fill), judgment (judge), multiple-choice (multi-choice), open-ended (open), and single-choice (single-choice). To simulate realistic noise, we inject 30% noisy samples into the training data through random perturbation and irrelevant distractors. We adopt a four-step iterative training procedure: (1) Model-1 is trained on a randomly sampled 25% of the full dataset,

from which 70% is used for training. (2) Model-2 selects 70% of a new 25% data slice (from the remaining 75%) using Model-1 as the evaluator. (3) Model-3 repeats this process using Model-2 as the evaluator on the next 25%. (4) Model-4 completes the final iteration using Model-3 as the evaluator on the remaining data. All models use Qwen2.5-7B as the base learner. Results are shown in Figure 9 and Figure 10, where Origin denotes the untrained Qwen2.5-7B baseline, and Iter-1 to Iter-4 correspond to the four models described above. Evaluation is performed on a held-out test set for each domain and question type. Case studies for each domain are provided in Appendix J. Implementation details are provided in Appendix A.3.

**Main Result.** As illustrated in Figure 10, model performance consistently improves across all domains, Anthropology, Economics, Law, and Philosophy, and all question types (fill-in-the-blank, judgment, multiple-choice, open-ended, single-choice) over the four iterations. Figure 9 further shows that the proportion of noisy samples in the selected training data steadily declines from Iter-1 to Iter-4 across all four domains demonstrating that LLMs can progressively learn to favor cleaner, more informative data through iterative self-selection. For instance, in the domain of Anthropology, the noise ratio drops from 30.00% in Iter-1 to 17.28% by Iter-4. This trend is consistent across domains, indicating an emergent capacity to filter out irrelevant or misleading content during the learning process. Compared to the untrained model baseline, which consistently selects data with 26% noise, trained models show a clear advantage in filtering out noise. This suggests that LLMs can learn to identify and prioritize higher-quality data through iterative training.

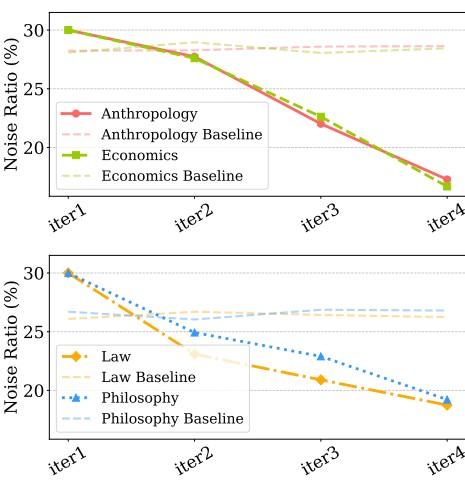

Figure 9: Noise ratios in data selected across four domains and iterations. Solid bars (e.g., Anthropology) show trained models; dashed bars (e.g., Anthropology Baseline) show untrained model selections.

Table 6: **Supervised fine-tuning hyper-parameters across domains.** Left: configuration for Mathematics, which uses a larger dataset and fewer epochs. Right: shared configuration for Philosophy, Anthropology, Economics, and Law.

| Math Domain | | Non-Math Domains | |
| --- | --- | --- | --- |
| **Hyper-parameter** | **Assignment** | **Hyper-parameter** | **Assignment** |
| Number of epochs | 2 | Number of epochs | 4 |
| Batch size per device | 1 | Batch size per device | 1 |
| Gradient accumulation steps | 4 | Gradient accumulation steps | 4 |
| Effective batch size | 32 | Effective batch size | 32 |
| Maximum sequence length | 2048 | Maximum sequence length | 2048 |
| Learning rate | 1e-5 | Learning rate | 1e-5 |
| Learning rate scheduler | cosine | Learning rate scheduler | cosine |
| Warm-up steps | 20 | Warm-up steps | 20 |
| Optimizer | AdamW | Optimizer | AdamW |
| Precision | FP16 | Precision | FP16 |

# I  Instruction Tuning Configuration

ts specifications of the instruction tuning corpora in each domain and Table 6 presents instruction tuning hyper-parameters. We conduct full-parameter supervised fine-tuning on five domain-specific corpora that cover Economics, Law, Philosophy, Anthropology, and Mathematics. Each corpus is built by pairing public domain texts with instruction–response examples produced through an automated prompt-and-verify pipeline: we first segment

source documents into short passages, then use GPT-4 to write an instruction for each passage and generate a reference answer, finally filtering out low-quality pairs with simple heuristic checks. After deduplication and shuffling, the resulting datasets contain between 12k and 100k samples per domain, as detailed in Table 7. We adopt the training recipe from LlamaFactory[5] and keep the same optimizer, scheduler, and precision across domains. Because the mathematics corpus is much larger than the others, we train for two epochs on Math and four epochs on the remaining domains while keeping the effective batch size, learning rate, sequence length, and other hyper-parameters identical (Table 6). All experiments are run on eight A100 80 GB GPUs with gradient accumulation to reach an effective batch size of 32.

## I.1 Task Descriptions and Token Length Statistics

Across the eight environments we track the number of generated tokens for every model–task pair and compare these budgets with the win-rate figures in Table 2. As shown in Table 8, token usage spans more than an order of magnitude: SpellingBee is consistently the cheapest task (all models $< 350$ tokens) whereas Tak is the most expensive (from 1.6 k for Llama-3.1-8B up to 6.6 k for Qwen3-8B). Model style has a larger impact than size: the Qwen3 series and Mistral models tend to answer verbosely, often emitting twice as many tokens as GPT-4o on the

Table 7: **Instruction–response corpora across five domains.**

| Domain | # Samples | Source |
|---|---|---|
| Economy | 12,408 | [65, 66] |
| Law | 29,648 | [61, 62] |
| Philosophy | 23,804 | [69, 70, 71] |
| Anthropology | 25,516 | [58] |
| Mathematics | 100,365 | [86] |

same input, while GPT-4o and GPT-4o-mini remain concise even on large boards. This verbosity does not buy success: GPT-4o secures the highest average win rate (0.75) while producing 30–50 % fewer tokens than its closest open-source competitors on most tasks. Conversely, Qwen3-8B generates the longest outputs yet ranks only mid-pack (0.49). Within individual tasks the pattern persists. On the compact SB game every model stays brief, but performance still ranges widely (0.10–0.70), showing that concision need not hurt accuracy. In the token-hungry TK environment only models that combine moderate length with strong heuristics—Mistral-7B (3.2 k tokens, win = 1.00) and Qwen3-14B (3.7 k, 1.00)—match GPT-4o's high score; models that answer either too tersely (Llama-3.1-8B, 1.6 k, 0.63) or too verbosely (Qwen2.5-7B, 5.6 k, 0.10) fall behind. Computing Pearson's $\rho$ across all 88 model–task pairs yields a weak negative correlation of roughly $-0.28$ between token count and win rate, indicating that quality of reasoning, not sheer output length, is the primary driver of performance in LearnArena.

Table 8: Average generated tokens per match for each model across the eight LearnArena environments.

| Model | CH | ST | TT | TD | SB | SM | TK | WC |
|---|---|---|---|---|---|---|---|---|
| **Llama-3.1-8B** | 635 | 1057 | 951 | 656 | 172 | 1004 | 1591 | 409 |
| **Mistral-7B-v0.3** | 886 | 1463 | 1900 | 1265 | 629 | 1095 | 3223 | 505 |
| **Mistral-8B-2410** | 663 | 1661 | 1977 | 772 | 541 | 1167 | 4149 | 289 |
| **Phi-4-mini** | 669 | 1222 | 1594 | 1315 | 344 | 1561 | 2111 | 296 |
| **Qwen2.5-14B-Chat** | 980 | 1370 | 2308 | 867 | 326 | 1206 | 4116 | 478 |
| **Qwen2.5-32B-Chat** | 551 | 1531 | 2252 | 1006 | 187 | 1413 | 3002 | 498 |
| **Qwen2.5-7B** | 567 | 1624 | 2259 | 1006 | 232 | 1147 | 5559 | 239 |
| **Qwen3-14B** | 2312 | 1037 | 4167 | 1949 | 2743 | 3938 | 3688 | 1396 |
| **Qwen3-8B** | 1648 | 1885 | 5736 | 2545 | 2030 | 3025 | 6577 | 1826 |
| **GPT-4o** | 1535 | 2110 | 1050 | 563 | 201 | 2021 | 4294 | 436 |
| **GPT-4o-mini** | 720 | 1349 | 952 | 531 | 172 | 1634 | 3607 | 405 |

---

[5]`https://github.com/hiyouga/LLaMA-Factory/blob/main/examples/train_full/llama3_full_sft_ds3.yaml`

## J    Case Studies of LfI in Non-Math Domains

The illustrative cases in Tables 9, Tables 10, Tables 11 and Tables 12 highlight the diversity and domain fidelity of the non-mathematical Learning-from-Instructor (LfI) benchmark. For each of the four domains—Anthropology, Economics, Law, and Philosophy—we present one exemplar of every question format used in our experiments: fill-in-the-blank, judgment, multiple-choice, open-ended, and single-choice. Each example is accompanied by a short answer and a concise explanation that reflects the expected reasoning chain. Together, these cases demonstrate the range of skills required by the benchmark: factual recall (fill), binary verification (judge), multi-step selection (multi-choice), free-form synthesis (open), and focused identification (single-choice). They also show that the prompts draw directly on domain-specific classics, ensuring that successful solutions depend on relevant expert knowledge rather than surface heuristics. This qualitative examination complements the quantitative results in Section H, confirming that the LfI tasks provide a realistic and varied testbed for evaluating how well large language models can identify, learn, and apply useful knowledge in the presence of noise.

Table 9: Question formats in anthropology domain.

**Multiple-Choice Question Example (multi-choice):**
*Question:* What are the two plans of government that have developed throughout human history?
*Choices:*
A: Social organization
B: Political organization
C: Religious organization
D: Economic organization
E: Military organization
*Answer:* A, B
*Explanation:* A and B are correct because the two primary plans of government that have developed throughout human history are social organization and political organization. Social organization, which is the older form, is based on kinship structures such as the gens, phratry, and tribe. Political organization, which emerged later, is based on territorial divisions such as townships, counties, and states. Religious, economic, and military organizations, while significant, are not listed as the two primary plans of government that have shaped societal structures.

**Judgment Question Example (judge):**
*Question:* The Aztec civilization extended slightly north of the Tropic of Cancer.
*Answer:* True
*Explanation:* This statement is true because the Aztec civilization, one of the most prominent in the New World, had its center in the region that extends just north of the Tropic of Cancer. This area, while still influenced by tropical conditions, provided a suitable environment for the Aztecs to develop their advanced society, including their complex social structures and agricultural practices.

**Single-Choice Question Example (single-choice):**
*Question:* What is one of the natural centers of subsistence mentioned in the text?
*Choices:*
A: The Mississippi Delta
B: The Great Plains
C: The Valley of the Columbia
D: The Appalachian Mountains
*Answer:* C
*Explanation:* C is correct because the Valley of the Columbia is mentioned as one of the natural centers of subsistence. This region is noted for its extraordinary variety and amount of subsistence resources, which made it an ideal place for the development of a surplus population and the subsequent formation of new tribes.

**Open-Ended Question Example (open):**
*Question:* How were the upper levels of Pueblo dwellings accessed?
*Answer:* By means of movable ladders and a hatchway in the roof
*Explanation:* The upper levels of Pueblo dwellings were reached using movable ladders and a hatchway in the roof. This design provided a means of access while also enhancing security, as the ladders could be removed to prevent unauthorized entry.

**Fill-in-the-Blank Question Example (fill):**
*Question:* In which region were the X-shaped bronze objects used as money, according to the passage?

*Answer:* Lunda
*Explanation:* The X-shaped bronze objects were used as money in the region of Lunda. These objects were made in Lunda and were widely accepted as currency throughout the Congo. The use of such distinctive and uniquely shaped metal objects as money highlights the diverse forms of currency that have been developed by different societies to facilitate trade and economic transactions.

Table 10: Question formats in economics domain.

**Multiple-Choice Question Example (multi-choice):**

*Question:* What does the text imply about the role of history in the study of law?

*Choices:*

A: History should be the exclusive focus of legal studies

B: History has no value in the study of law

C: History should be integrated with other methods in legal studies

D: History should be used to pursue pure abstractions

E: History should be used to understand the evolution of legal principles and practices

*Answer:* C, E

*Explanation:* C and E are correct because history should be integrated with other methods in legal studies and should be used to understand the evolution of legal principles and practices. History should not be the exclusive focus of legal studies, does have value in the study of law, and should not be used to pursue pure abstractions.

**Judgment Question Example (judge):**

*Question:* During the middle ages, candles were very expensive, costing about 1-1/3 to 2 shillings per pound.

*Answer:* True

*Explanation:* Historical records indicate that candles were a luxury item during the middle ages, with prices ranging from 1-1/3 to 2 shillings per pound. This high cost made candles an expensive commodity, accessible primarily to the wealthy, and contributed to their status as a symbol of affluence and status.

**Single-Choice Question Example (single-choice):**

*Question:* Which Roman emperor made a significant legal change concerning the treatment of abandoned children?

*Choices:*

A: Julius Caesar

B: Augustus

C: Constantine the Great

D: Nero

*Answer:* C

*Explanation:* C is correct because Constantine the Great, in the year 315 AD, made a significant legal change concerning the treatment of abandoned children. This change likely aimed to improve the conditions and legal standing of abandoned children, reflecting a shift in Roman societal values and legal practices.

**Open-Ended Question Example (open):**

*Question:* What is the relationship between the development of private property and the advancement of civilization?

*Answer:* Private property has developed as civilization and well-being have advanced

*Explanation:* The evolution of private property is closely linked to the progress of civilization and the improvement of living standards. As societies become more developed and culturally advanced, the concept and institution of private property become more established, reflecting and contributing to the overall well-being and prosperity of the community.

**Fill-in-the-Blank Question Example (fill):**

*Question:* The unnecessary destruction of wealth, such as spending 10,000 dollars on fireworks, is considered _______, regardless of whether the money remains in the country.

*Answer:* disastrous

*Explanation:* Economic theory emphasizes the importance of productive and sustainable use of resources. When wealth is spent on non-productive activities, such as extravagant fireworks displays, it is considered a wasteful and disastrous use of resources. This is because the destruction of wealth, even if the money circulates within the country, does not contribute to long-term economic growth or development. The resources used for such displays could have been allocated to more productive uses, such as investment in infrastructure or education.

Table 11: Question formats in law domain.

---

**Multiple-Choice Question Example (multi-choice):**
*Question:* Which of the following authors wrote treatises that were used as textbooks for teaching International Law?
*Choices:*
A: William Edward Hall
B: Sir Edward Shepherd Creasy
C: Theodore D. Woolsey
D: Heinrich Bernhard Oppenheim
E: Luigi Casanova
F: Ernest Nys
*Answer:* A, B, C, D, E
*Explanation:* A, B, C, D, and E are correct because William Edward Hall, Sir Edward Shepherd Creasy, Theodore D. Woolsey, Heinrich Bernhard Oppenheim, and Luigi Casanova all wrote treatises that were used as textbooks for teaching International Law. These works were designed to provide comprehensive and accessible introductions to the field, suitable for students and practitioners. Ernest Nys, while a significant author, did not specifically write a textbook.

---

**Judgment Question Example (judge):**
*Question:* Private enemy property on a belligerent's territory can be confiscated according to the customary rule of International Law.
*Answer:* False
*Explanation:* The customary rule of International Law now prohibits the confiscation of private enemy property on the territory of a belligerent. This rule evolved through international treaties, municipal laws, and decrees, making the old practice of confiscation obsolete.

---

**Single-Choice Question Example (single-choice):**
*Question:* What is the significance of the Hague Conferences of 1899 and 1907 in the context of guerilla warfare?
*Choices:*
A: They banned all forms of guerilla warfare
B: They provided a legal framework for the treatment of guerilla fighters
C: They recognized guerilla warfare as a legitimate form of war
D: They established the rules for the treatment of prisoners of war
*Answer:* B
*Explanation:* B is correct because the Hague Conferences of 1899 and 1907 provided a legal framework for the treatment of guerilla fighters. These conferences established conditions under which guerilla fighters could be treated as soldiers, thus providing them with certain protections under international law.

---

**Open-Ended Question Example (open):**
*Question:* What is the purpose of pledging movable property in the context of international treaties?
*Answer:* To secure the performance of a treaty
*Explanation:* Pledging movable property in international treaties serves as a form of collateral to ensure that the obligations outlined in the treaty are fulfilled. This practice acts as a safeguard, where the pledged items can be retained or used to compensate the other party if the treaty obligations are not met.

---

**Fill-in-the-Blank Question Example (fill):**
*Question:* The concept of prescription in international law is not solely based on the _______ of time.
*Answer:* passage
*Explanation:* Prescription in international law is not determined by a fixed period of time alone. It involves a combination of factors, including the stability of possession, the absence of protests, and the general acceptance by the international community. The passage of time is only one aspect of the broader context that contributes to the formation of a prescriptive title.

---

Table 12: Question formats in philosophy domain.

**Multiple-Choice Question Example (multi-choice):**
*Question:* Which of the following philosophers is praised for his deeper insights compared to Rousseau?
*Choices:*
A: Voltaire
B: Schopenhauer
C: Locke
D: Byron
E: Herodotus
F: Plutarch
*Answer:* B, C
*Explanation:* B and C are correct because the unnamed great man, likely Schopenhauer, is praised for his depth of thinking, which includes insights aligned with Locke's principle. Voltaire, Byron, Herodotus, and Plutarch are not mentioned as being praised for deeper insights compared to Rousseau.

**Judgment Question Example (judge):**
*Question:* Satisfaction of a wish can lead to the creation of a new wish.
*Answer:* True
*Explanation:* Satisfaction of a wish often leads to the creation of a new wish because desires and wants are inherently dynamic and ever-evolving. Once a particular goal is achieved, the human mind tends to focus on the next goal, creating a continuous cycle of desire and fulfillment. This is rooted in the nature of human psychology, where the satisfaction of one need or desire often reveals or intensifies another, keeping individuals in a perpetual state of seeking.

**Single-Choice Question Example (single-choice):**
*Question:* Which of the following are considered essential fundamental constituent elements of language according to the passage?
*Choices:*
A: Noun, verb, and adjective
B: Pronoun, article, and adverb
C: Substantive, adjective, and verb
D: Substantive, article, and pronoun
*Answer:* C
*Explanation:* C is correct because the essential fundamental constituent elements of language, as mentioned, are substantive (which is another term for noun), adjective, and verb. These elements are crucial for the structure and function of language, providing the core components needed to form sentences and convey meaning.

**Open-Ended Question Example (open):**
*Question:* What is the first step towards achieving a metaphysical understanding of the world, according to the text?
*Answer:* Bringing to distinct consciousness and firmly retaining the difference between physics and metaphysics.
*Explanation:* The first step in moving towards a metaphysical understanding is to clearly recognize and maintain the distinction between physics and metaphysics. Physics deals with the observable and measurable aspects of the world, while metaphysics explores the deeper, underlying nature of reality. Understanding this difference is crucial for recognizing the limitations of physical explanations and the need for a metaphysical approach.

**Fill-in-the-Blank Question Example (fill):**
*Question:* The aesthetic effect of architectural works is in exact proportion to the _______ of the building.
*Answer:* size
*Explanation:* The aesthetic impact of a building increases with its size. Larger structures are more effective in showcasing the forces of nature, such as gravity, in a more apparent and impressive manner. This is because larger masses make the action of these forces more visible and tangible, enhancing the overall aesthetic experience. Therefore, the size of a building is a critical factor in its aesthetic value.

## K  Prompt Design for LearnArena

To support the evaluation of general learning ability in LearnArena, we design three types of prompts corresponding to the three learning dimensions: Concept, Instructor, and Experience. Each prompt is carefully constructed to elicit behavior aligned with the intended mode of learning, while remaining concise and task-relevant. For Learning from Concept, in Table 13, the prompt introduces the game setting and provides a concise, structured request for strategic advice. The system prompt frames the LLM as an expert strategist, while the user prompt asks for a list of key principles, tactics, and pitfalls, grounded in the rules of the game. This setup encourages the model to abstract general strategies directly from conceptual understanding. For Learning from Instructor, in Table 14, the prompt simulates feedback-based instruction. The system prompt positions the model as a game instructor giving targeted feedback. The user provides a summary of game history from prior rounds, and the model is asked to analyze and respond in a structured format. This prompt encourages the model to extract lessons from explicit advice and past decisions, simulating instructor-led reflection. For Learning from Experience, in Table 15, the prompt shifts the role to that of a self-reflective player. The system prompt instructs the model to analyze its own performance and derive improvements. The user supplies a history of recent gameplay, and the model is prompted to identify useful patterns, good and bad moves, and general strategic takeaways. This design promotes autonomous learning from accumulated experience. All prompts share a consistent structure and tone, enabling fair comparison across models and environments. By isolating each learning mode, we can evaluate how well a model generalizes knowledge from rules, instruction, and prior experience independently.

Table 13: Prompt template for the concept learning setting.

---

**Prompt for Concept**

**System Prompt:**
You are an expert game strategist with deep knowledge of game theory and optimal play. Your task is to provide concise, actionable strategic advice that will help a player win. Focus on identifying winning patterns, key decision points, and optimal strategies.

**User Prompt:**
For this game '{game}', provide brief winning strategies based on this initial observation.

WINNING STRATEGIES:

1. Top 3–5 strategic principles that lead to victory

2. Best opening moves or early game tactics

3. Key patterns to recognize during gameplay

4. Critical mistakes to avoid

**Rules of the Game**: {rules}

Keep your response concise and focused on practical advice that will maximize winning chances.

---

## L  Prompt Design for the Passive/Interactive Setting

To support our experiments in Section 4.1, we design distinct prompting strategies to simulate the two instruction-based learning paradigms: Passive Consumption and Interactive Clarification. These prompts define how the instructor and learner interact during the generation and clarification phases of training data construction. In the Passive setting, the instructor model is prompted to solve a math problem directly, using a system instruction that encourages detailed step-by-step reasoning and a final answer. The learner is then trained solely on these final outputs without any follow-up interaction. The corresponding prompt template is shown in Table 17. In the Interactive setting, the instructor-learner interaction is decomposed into a three-step process. First, the instructor provides a

Table 14: Prompt template for the instructor learning setting.

| Prompt for Instructor |
| --- |
| **System Prompt:**
You are an expert game instructor. Your task is to provide detailed strategic advice for the game based on the provided information.

**User Prompt:**
Analyze this history and provide strategic advice.

**Game history:** {game history}

Provide your analysis in this format:
    1. Strategic principles to follow
    2. Specific moves to consider
    3. Moves to avoid
    4. Key patterns to watch for |

Table 15: Prompt template for the experience learning setting.

| Prompt for Experience |
| --- |
| **System Prompt:**
You are a player analyzing your own gameplay. Your task is to identify strengths, weaknesses, and learn from this experience to enhance future performance.

**User Prompt:**
Analyze this history and provide strategic advice.

**Game history:** {game history}

Provide your analysis in this format:
    1. Strategic principles to follow
    2. Specific moves to consider
    3. Moves to avoid
    4. Key patterns to watch for |

detailed solution to a math problem, identical to the passive setting. Second, the learner (student) reviews the solution and is prompted to identify specific confusions or follow-up questions based on the instructor's response. Finally, the instructor receives the student's clarifying question and provides a targeted explanation aimed at resolving the confusion. This interactive exchange is used as training data for the learner model, capturing a more realistic and adaptive instructional process. This procedure reflects a more realistic and adaptive instructional scenario. The full prompting sequence for this setting is shown in Table 16.

We also present a concrete case study illustrating the benefits of the Interactive prompting approach, as shown in Table 19, Table 20, and Table 21. In this example, the learner poses meaningful and specific questions about the instructor's initial solution, such as clarifying the strategy behind number grouping, order of operations, and the existence of alternative solutions. The instructor's detailed responses systematically address these queries, clearly enhancing the instructional quality compared to the passive setting. This exchange highlights the distinct advantage of interactive clarification,

where learner-generated questions provoke richer, more targeted explanations, ultimately improving learning effectiveness.

Table 16: Interactive prompts for mathematical domain.

---

**Interactive Step 1: Instructor Receives Problem**

**System Prompt:**
You are a mathematician and educator. Solve the following math problem with accurate, complete, and clear explanations. Break down your reasoning into a logical chain of steps, and provide the final answer only after completing the reasoning.

**User Prompt:**
{question}

---

**Interactive Step 2: Student Reviews Solution**

**System Prompt:**
You are a math student. You will read the math problem and a solution provided by your math teacher. If you have any questions or confusions about the solution, ask specific questions about those parts.

**User Prompt:**
Math Problem: {question}
Here is the solution provided by your teacher: {answer}
If you have any questions or confusions about the solution, please list them.

---

**Interactive Step 3: Instructor Addresses Confusions**

**System Prompt:**
You are a mathematician and educator dedicated to resolving confusions about math solutions. Provide clear, step-by-step explanations to logically address each confusion.

**User Prompt Context:**
**Initial Question:** {question}

**Provided Solution:** {answer}

**Student's Confusions:** Confusions about the solution: {confusion}

Please address the confusions.

---

Table 17: Passive prompt for mathematical domain.

Table 18: Case study: passive prompt in mathematics.

**Passive Prompt Math Problem and Solution:**

*Prompt:* You are a mathematician and educator. Solve the following math problem with accurate, complete, and clear explanations. Break down your reasoning into a logical chain of steps, and provide the final answer only after completing the reasoning.

*Question:* If $a$, $b$, and $c$ are positive integers satisfying $ab + c = bc + a = ac + b = 41$, what is the value of $a + b + c$?

*Solution (Output):* We have $ab + c = 41$ and $bc + a = 41$, so $ab + c = bc + a$. Subtracting the two equations, we have $ab - bc + c - a = 0 \Rightarrow b(a - c) + (c - a) = 0 \Rightarrow (b - 1)(a - c) = 0$. Therefore, $a = c$ or $b = 1$.

If $a = c$, we have $ab + c = a^2 + b = 41$. We also have $bc + a = ab + c = 41$. If $a^2 + b = 41$, then $a^2 \leq 41$, so $a \leq 6$. If $a = 6$, then $36 + b = 41 \Rightarrow b = 5$. If $a = c = 6$ and $b = 5$, then our three equations are satisfied, so $a + b + c = 6 + 5 + 6 = \boxed{17}$. If $a < 6$, then $b \geq 6$, so $b > a$. We also have $a^2 + b = 41$, so $b = 41 - a^2 < 25$. Therefore, $6 \leq a < 6$, so $a = 6$, and we have the same solution as before.

If $b = 1$, we have $ab + c = a + c = 41$ and $bc + a = c + a = 41$. Therefore, $a + c = c + a$, which is always true. However, we also have $ac + b = ac + 1 = 41 \Rightarrow ac = 40$. Therefore, $a$ and $c$ are factors of 40. The only factor pairs of 40 are $(1, 40)$, $(2, 20)$, $(4, 10)$, and $(5, 8)$. However, we are looking for positive integers $a, b, c$ such that $a + c = 41$, so the only possible solution is $a = 5$ and $c = 8$. Therefore, $a + b + c = 5 + 1 + 8 = \boxed{14}$. The answer is 14

Table 19: Case study: interactive prompts for mathematical domain.

**Interactive Step 1: Instructor Receives Problem**

**System Prompt:**
You are a mathematician and educator. Solve the following math problem with accurate, complete, and clear explanations. Break down your reasoning into a logical chain of steps, and provide the final answer only after completing the reasoning.

**User Prompt (Question):**
Using the numbers 1, 2, 3, 4, 5, 6, 7, 8, 9 in order, and only using the operations of addition and subtraction, how can you make the expression equal 100? Parentheses can be used to change the order of operations.

Table 20: Case study: interactive prompts for mathematical domain.

---

**Interactive Step 2: Student Reviews Solution & Asks Questions**

---

## System Prompt:
You are a math student. You will read the math problem and a solution provided by your math teacher. If you have any questions or confusions about the solution, ask specific questions about those parts.

## User Prompt:
**Math Problem:** Using the numbers 1, 2, 3, 4, 5, 6, 7, 8, 9 in order, and only using the operations of addition and subtraction, how can you make the expression equal 100? Parentheses can be used to change the order of operations.

**Here is the solution provided by your teacher (Initial Answer):** To solve the problem of making the expression using the numbers 1 through 9 in order, and only using addition and subtraction to equal 100, we need to strategically place the operations and parentheses. Let's break down the solution step-by-step.

First, let's consider the structure of the problem. We need to use all the digits from 1 to 9 in order and use only addition and subtraction to reach 100. We will experiment with different placements of operations and parentheses to find a valid expression.

One effective approach is to group numbers in a way that allows us to use the operations to balance the result to 100. Let's start by trying to use the larger numbers to get close to 100 and then adjust with smaller numbers.

Consider the sequence:
$$123 - 45 - 67 + 89$$

Let's break this down: 1. Calculate $123 - 45$:
$$123 - 45 = 78$$

2. Next, calculate $78 - 67$:
$$78 - 67 = 11$$

3. Finally, add 89 to 11:
$$11 + 89 = 100$$

So, the expression $123 - 45 - 67 + 89$ equals 100.
Therefore, the final answer is:

$$\boxed{123 - 45 - 67 + 89}$$

**If you have any questions or confusions about the solution, please list them (Student's Confusions):** I understand the solution provided, but I have a few questions and points of confusion:

1. **Grouping Numbers**: Why did we choose to group the numbers as 123, 45, 67, and 89? Is there a specific strategy or intuition behind this grouping?

2. **Order of Operations**: How do we ensure that the order of operations (addition and subtraction) is correctly applied when evaluating the expression? Specifically, how do the parentheses ensure the correct order?

3. **Alternative Solutions**: Are there other ways to group the numbers and use addition and subtraction to reach 100? If so, could you provide an example of another valid expression?

4. **General Strategy**: Is there a general method or algorithm to solve problems like this, where we need to use a sequence of numbers and operations to reach a specific target?

5. **Verification**: How can we verify that no other combination of addition and subtraction using the numbers 1 through 9 in order will also result in 100?

---

Table 21: Case study: interactive prompts for mathematical domain — Part 1

---

**Interactive Step 3: Instructor Addresses Confusions**

---

**System Prompt:**
You are a mathematician and educator dedicated to resolving confusions about math solutions.
Provide clear, step-by-step explanations to logically address each confusion.

**User Prompt Context:**
*Initial Question:* Using the numbers 1, 2, 3, 4, 5, 6, 7, 8, 9 in order, and only using the operations of addition and
subtraction, how can you make the expression equal 100? Parentheses can be used to change the order of operations.
*Provided Solution (Initial Answer):*
To solve the problem of making the expression using the numbers 1 through 9 in order, and only using addition and subtraction
to equal 100, we need to strategically place the operations and parentheses. Let's break down the solution step-by-step.
First, let's consider the structure of the problem. We need to use all the digits from 1 to 9 in order and use only addition and
subtraction to reach 100. We will experiment with different placements of operations and parentheses to find a valid expression.
One effective approach is to group numbers in a way that allows us to use the operations to balance the result to 100. Let's
start by trying to use the larger numbers to get close to 100 and then adjust with smaller numbers.

Consider the sequence:

$$123 - 45 - 67 + 89$$

1. Calculate $123 - 45 = 78$
2. Then $78 - 67 = 11$
3. Finally $11 + 89 = 100$

Therefore the expression $123 - 45 - 67 + 89$ equals 100, and the final answer is $\boxed{123 - 45 - 67 + 89}$.

Table 21: Case study: interactive prompts for mathematical domain — Part 2

---

**Student's Confusions:**

1. The strategy behind grouping $123, 45, 67, 89$.
2. Ensuring the order of operations, especially the role of parentheses.
3. Whether alternative solutions exist and an example.
4. Whether a general algorithm exists for such problems.
5. How to verify that no other combinations yield 100.

---

**Instructor's Response:**

**1. Grouping Numbers (**$123 \mid 45 \mid 67 \mid 89$**).**
A reliable heuristic is to fuse the early digits into the largest possible chunk (here 123) so you begin near the target of 100.
Next, look for a pair of medium-sized numbers whose difference is small (45 and 67 differ by 22) and a final adjustment
term ($\pm 89$) that brings the running total to the goal. This "large-number-first, cancel-with-a-pair, fine-tune" pattern
dramatically shrinks the search space compared with blind trial-and-error.

**2. Order of Operations & Parentheses.**
Because addition and subtraction are left-associative and equal-priority, `123 - 45 - 67 + 89` is automatically evaluated
left-to-right, producing 100 without extra parentheses. Parentheses matter only when we deliberately override that rule—
e.g. ( $123 - (45 - 67) + 89$ ), which forces $45 - 67$ first.

**3. Alternative Solutions.**
Multiple expressions satisfy the constraints, for example
- $123 + 4 - 5 + 67 - 89 = 100$
- $12 + 3 - 4 + 5 + 6 + 78 = 100$

Each follows the same philosophy: create or destroy "mass" with big blocks, then make small tweaks.

**4. General Strategy / Algorithm.**
A systematic method is depth-first search with pruning:
- At each digit boundary, choose to concatenate or to insert $+ / -$.
- Track a running total; abandon any branch whose partial total already overshoots a sensible bound.
- There are $2^8 = 256$ operator patterns and $2^8 = 256$ digit-merge patterns, so the full tree has at most 65 536 leaves—
  small for a computer and still manageable by hand with pruning.

**5. Verifying Uniqueness.**
Write a short program (or spreadsheet) that enumerates every merge / operator pattern; collect the ones equalling 100.
You will find only a handful of distinct families and no hidden outliers, giving a rigorous guarantee that you have all solutions.
Armed with these tactics—large-chunk heuristics, operator-precedence insights, a DFS-with-pruning template, and programmatic
verification—you can tackle not only this puzzle but the entire class of "insert $\pm$ to hit a target" challenges with confidence.

---

```
[GAME] You are Player 0 in Tak.
Your goal is to connect two opposite edges of the board with your pieces to form a road...
% --- Rules and examples have been abridged for brevity ---

[Player 0] Given that the board is currently empty, I will begin by placing a flat stone...
[place () {(0,0): [F0]}]

[GAME] Player 0 placed a piece on ([(0, 0)]). New board state:
        0        1        2        3
     -----------------------------------
  0 | (1) F0 |        |        |        |
     -----------------------------------
  1 |        |        |        |        |
     -----------------------------------
  2 |        |        |        |        |
     -----------------------------------
  3 |        |        |        |        |
     -----------------------------------
[Player 1] For my move, I will place a wall stone to block potential pathways...
[place () {(0,1): [W1]}]

[GAME] Player 1 placed a piece on ([(0, 1)]). New board state:
        0        1        2        3
     -----------------------------------
  0 | (1) F0 | (1) W1 |        |        |
     -----------------------------------
...
% --- Intermediate moves are omitted for brevity ---
...

[GAME] Player 1 placed a piece on ([(3, 1)]). New board state:
        0        1        2        3
     -----------------------------------
  0 | (1) F0 | (1) W1 | (1) F0 |        |
     -----------------------------------
  1 | (1) F0 | (1) W1 | (1) W1 |        |
     -----------------------------------
  2 | (1) F0 | (1) W1 | (1) F0 |        |
     -----------------------------------
  3 |        | (1) W1 |        |        |
     -----------------------------------
```

Figure 11: Abridged game history log as presented to the agent. It includes the initial rules, followed by a turn-by-turn log of actions and resulting board states.

