# OpenReview forum: "Unveiling the Learning Mind of Language Models: A Cognitive Framework and Empirical Study"
_NeurIPS.cc/2025/Conference — NeurIPS 2025 poster_

### Official Review · Reviewer_932f · 2025-06-08

**Clarity:** 3
**Significance:** 3
**Originality:** 3
**Rating:** 4
**Confidence:** 2

**Summary:**

The paper introduces a framework, inspired by cognitive psychology, to evaluate the ability of LLMs to acquire new knowledge and abilities (learning ability). They conduct an empirical study to evaluate the learning ability of existing LLMs and introduce a benchmark to quantify this.

**Questions:**

None.

**Ethical Concerns:**

["NO or VERY MINOR ethics concerns only"]

**Final Justification:**

Q1 of weakness -- mostly addressed

Q2 of weakness -- partially addressed, they should add a limitation of scope for using the term "concept" in the discussion section

Q1 of limitations -- mostly addressed

Q2 of limitations -- mostly addressed

Overall, the authors put effort into writing and experiments for addressing the weaknesses and limitations I mentioned. I think including these in the paper will improve it.

**Limitations:**

Limitations:
1. Evaluation scope of only using Qwen models
2. Evaluation scope of the various benchmarks in Math, games, etc.

**Quality:**

2

**Strengths And Weaknesses:**

Strengths:
1. Overall, the paper provides a good contribution toward understanding whether and how LLMs can adaptively learn.
2. I like the breakdown into learning from an instructor, concept, and experience.

Weaknesses:
1. It would be interesting to see more elaboration on how the framework is rooted in cognitive psychology, and also compare the LLM result trends in this paper with human results in existing psychology papers. (Here, I think word explanations are sufficient, not asking for additional experiments.)
2. Providing rules and strategies for playing games as a proxy for "Concept" seems slightly narrow or weak. The comparison here is whether an LLM can play a game with versus without the rules or strategies provided, which seems not the most fitting to the notion of "concept". The definition of "concept" and how humans learn concepts has been studied in cognitive psychology, and it would be nice to evaluate if LLMs can, similarly to humans, also learn an (abstract) concept and apply it to new scenarios.

---

> ### Author Rebuttal · Authors · 2025-07-31
>
> Dear Reviewer 932f,
>
> Thank you so much for your valuable feedback!
>
> 1. Reply to Q1 in weakness.
>
>    ---
>
>    Our framework is grounded in long-standing perspectives from cognitive science and educational theory (e.g., [1,2,3]), which emphasize that human learning emerges through multiple complementary pathways. Drawing from these foundations, we propose a tripartite decomposition of learning ability into three cognitively motivated dimensions:
>
>    - Learning from Instructor (LfI): (e.g., [4, 5]). Here, the key is the presence of an *interactive teaching agent*—the instructor—that can shape model behavior.
>    - Learning from Concept (LfC): (e.g., [6]). This dimension focuses on how models use declarative conceptual knowledge to generalize across tasks.
>    - Learning from Experience (LfE):  (e.g., [7,8]). learning from past interactions, e.g., previous inputs, outputs, or trajectories, without explicit instruction
>
>    Interestingly, some observed trends in our study parallel known human learning patterns—for example, the scale-emergent use of concept inputs mirrors how abstract reasoning improves with cognitive capacity [9], and the many-shot degradation resembles overload effects in human memory systems [10].
>
>    We will clarify these connections in the revision to better highlight the cognitive grounding of our framework.
>
>    ---
>
> 2. Reply to Q2 in weakness.
>
>    ---
>
>    Thank you for the insightful comment. We agree that the notion of concept in cognitive psychology is rich and multifaceted, often extending beyond explicit rules or strategies to include structured abstraction, category formation, and transfer to novel domains.
>
>    In our work, we adopt a pragmatic and operational definition of “concept” tailored to the current capabilities of LLMs. Specifically, we focus on concise, distilled abstractions (e.g., rules, heuristics, symbolic patterns) that serve as testable approximations of conceptual input. This makes it possible to evaluate whether a model can internalize and apply high-level structure, even without explicit training.
>
>    While our setup centers on rule-based environments, this is not meant to reduce the broader meaning of "concept." Instead, we view it as a first step toward measuring concept-driven generalization in LLMs under controlled conditions. In future work, we hope to extend this to more open-ended or human-like concept learning tasks.
>
>    In short, although the term "concept" is shared, the operational scope differs, our use is intentionally narrow for the sake of empirical tractability, and we appreciate the reviewer’s point in encouraging a deeper connection to cognitive theory. We will clarify this distinction in the revision.
>
>    ---
>
> 3. Reply to Q1 in Limitations.
>
>    ---
>
>    Thank you for the question. We use Qwen models extensively because the family includes a **broad range of scales (1.5B to 72B)** under a consistent architecture and training regime. This makes it uniquely well-suited for **analyzing scaling effects** across different learning dimensions (e.g., Fig. 2–3, 4, 5, 6).
>
>    That said, we also include **other model families**, such as GPT-4o, LLaMA3, Mistral, and Phi-3—in multiple parts of the study (e.g., Sec. 4.1, Fig. 2, Sec. 6.2, Sec. 7). These comparisons show consistent trends, supporting the **robustness and generality** of our conclusions beyond Qwen.
>
>
>
>    We agree that using a single fixed opponent (Qwen2.5-32B) may raise concerns about overfitting to its strategies.
>
>    To address this, we are currently extending our evaluation by including different Player-0:
>
>    Player0 = Phi-4
>
>    | Player1      | CH   | ST   | SB   | SM   |
>    | ------------ | ---- | ---- | ---- | ---- |
>    | gpt-4o       | 1.00 | 0.90 | 0.65 | 1.00 |
>    | gpt-4o-mini  | 0.65 | 0.88 | 0.86 | 0.72 |
>    | llama-3.1-8b | 0.65 | 0.70 | 0.50 | 0.75 |
>    | mistral-8b   | 0.40 | 0.75 | 0.28 | 0.69 |
>    | qwen2.5-7b   | 0.70 | 0.75 | 0.35 | 0.35 |
>    | qwen3-8b     | 0.75 | 0.65 | 0.50 | 0.70 |
>
>
>
>    Player0 = GPT4-o-mini
>
>    | Player1      | CH   | ST   | SB   | SM   |
>    | ------------ | ---- | ---- | ---- | ---- |
>    | gpt-4o       | 0.80 | 1.00 | 0.65 | 1.00 |
>    | gpt-4o-mini  | 0.50 | 0.55 | 0.45 | 0.55 |
>    | llama-3.1-8b | 0.25 | 0.85 | 0.10 | 0.70 |
>    | mistral-8b   | 0.15 | 0.85 | 0.20 | 0.45 |
>    | qwen2.5-7b   | 0.45 | 0.70 | 0.25 | 0.35 |
>    | qwen3-8b     | 0.40 | 0.95 | 0.55 | 0.50 |
>
>
>
>    Player0 = GPT4-o
>
>    | Player1      | CH   | ST   | SB   | SM   |
>    | ------------ | ---- | ---- | ---- | ---- |
>    | gpt-4o       | 0.60 | 0.50 | 0.45 | 0.50 |
>    | gpt-4o-mini  | 0.15 | 0.45 | 0.35 | 0.10 |
>    | llama-3.1-8b | 0.00 | 0.50 | 0.20 | 0.15 |
>    | mistral-8b   | 0.00 | 0.43 | 0.05 | 0.10 |
>    | qwen2.5-7b   | 0.25 | 0.53 | 0.05 | 0.05 |
>    | qwen3-8b     | 0.45 | 0.65 | 0.15 | 0.05 |
>
>    We agree that relying on a single Player-0 (Qwen2.5-32B) may raise concerns about overfitting to its strategies. To address this, we conducted preliminary experiments using alternative Player-0 models: **Phi-4**, **GPT-4o-mini**, and **GPT-4o**. For each Player-0, we computed the average performance of each Player-1 across four environments (CH, ST, SB, SM), ranked the models accordingly, and calculated the **Spearman rank correlation** with the original ranking under Qwen2.5-32B.
>
>    - Player-0 = **Phi-4** → ρ = **1.000**
>    - Player-0 = **GPT-4o-mini** → ρ = **0.829**
>    - Player-0 = **GPT-4o** → ρ = **0.714**
>
>    These results show that **model rankings remain largely consistent across different opponents**, suggesting that our benchmark findings are not overly sensitive to the choice of Player-0. We will include this analysis in the revised Appendix.
>
>    Thank you again for the helpful suggestion!
>
>    ---
>
> 4. Reply to Q2 in Limitations.
>
>    ---
>
>    Thank you for the question. Could you kindly elaborate a bit more on this point?
>
>    ---
>
> Once again, we appreciate your valuable feedback and the opportunity to address your concerns.
>
> **Reference:**
>
> [1] The modularity of mind
>
> [2] Experience as the source of learning and development
>
> [3] Putting students on the path to learning: The case for fully guided instruction
>
> [4] Rapid instructed task learning: A new window into the human brain's unique capacity for flexible cognitive control
>
> [5] Rapid instruction-based task learning (RITL) in schizophrenia
>
> [6] An Alternative to Cognitivism: Computational Phenomenology for Deep Learning
>
> [7] Experiential learning: Experience as the source of learning and development
>
> [8] LLM Agents Are Experiential Learners
>
> [9] Reasoning ability is (little more than) working-memory capacity?!
>
> [10] The magical number 4 in short-term memory: A reconsideration of mental storage capacity

---

> > ### Comment · Reviewer_932f · 2025-08-04
> >
> > Q2 of Limitations -- my point was mostly that evaluating math and games etc. is just one domain and does not cover satisfactorily the range of the terms the paper has used, e.g., "concept". But I see your answer in reply to Q1 of weaknesses, and I agree that this is challenging. I think that it would be sufficient for you to mention in the discussion section of the paper that the scope should be further expanded in future work. I think that I do not need any additional clarification.
> >
> > Overall, the authors put effort into writing and experiments for addressing the weaknesses and limitations I mentioned. I think including these in the paper will improve it.

---

> > > ### Author Response · Authors · 2025-08-05
> > >
> > > Dear Reviewer 932f,
> > >
> > > Thank you for your follow-up and helpful suggestions.
> > > We agree with you, and will add the suggested content in the revision. For Q2, we will add a note in the limitations section.
> > >
> > > Best regards!

---

> ### Author Response · Authors · 2025-08-04
>
> Dear Reviewer 932f,
>
> Thank you again for your valuable feedback on our submission. We hope that our responses have addressed your questions and concerns.
> If you have further question, we hope to take the last chance in the rest days to clarify.
>
> Best regards!

---

### Official Review · Reviewer_vwyb · 2025-07-01

**Clarity:** 2
**Significance:** 2
**Originality:** 2
**Rating:** 4
**Confidence:** 3

**Summary:**

This paper introduces a cognitively based framework to evaluate the learning abilities of LLMs. This framework decomposes general learning into three dimensions: Learning from Instructor, Learning from Concept, and Learning from Experience. The authors conduct comprehensive empirical studies across these dimensions and introduce LearnArena, a unified benchmark to assess the learning capabilities of LLMs.

**Questions:**

- How do the authors differentiate whether the tasks are measuring genuine learning rather than improved performance due to from better input formatting?
- The authors suggest that LLMs struggle with many-shot learning is essential, but could you provide a more systematic analysis of where and why this breakdown occurs?

**Ethical Concerns:**

["NO or VERY MINOR ethics concerns only"]

**Final Justification:**

The additional experiments and anlaysis provided by the authors in the rebuttal address my major concerns.

**Limitations:**

Yes

**Quality:**

2

**Strengths And Weaknesses:**

**Strengths:**
- The paper examines an unexplored gap in current LLM evaluation: most existing benchmarks (MMLU, BIG-Bench, HELM) focus on static knowledge demonstration rather than learning dynamics.
- The authors provide three key empirical findings : (1) Interactive learning is consistently superior; (2) Conceptual understanding depends on model scale; (3) Many-shot learning paradox.
- The paper is well-structured, with clear explanations of methodology and experiments. The results are presented clearly through informative visualizations.

**Weaknesses:**
- While the three-dimensional framework is intuitive, the paper lacks a formal analysis of how these dimensions interact or whether they are truly orthogonal. The claim that LLMs "learn" in these settings, as opposed to simply performing better with different inputs, could be more rigorously examined.
- While the paper successfully documents various learning phenomena across the three dimensions, it lacks sufficient analysis to explain the underlying causes of observed behaviors.
- The LearnArena benchmark relies on a single, fixed opponent (Player-0, specifically Qwen2.5-32B) for all evaluations. This creates a risk that evaluated models could achieve high scores by overfitting to the specific strategies and weaknesses of this one opponent, rather than by developing truly generalizable learning abilities. Furthermore, performance in LearnArena is measured solely by the average win rate. This metric, while intuitive, may not capture important nuances of the learning process. For example, a model could lose a game but still demonstrate superior adaptation or better adherence to instructions.

---

> ### Author Rebuttal · Authors · 2025-07-31
>
> Dear Reviewer vwyb,
>
> Thank you so much for your valuable feedback!
>
> 1. Reply to Q1 in weakness.
>
>    ---
>
>    Thank you for the thoughtful question. We address both parts below:
>
>    **(1) Why we propose these three dimensions**Our framework is grounded in long-standing perspectives from cognitive science and educational theory (e.g., [1,2,3]), which emphasize that human learning emerges through multiple complementary pathways. Drawing from these foundations, we propose a tripartite decomposition of learning ability into three cognitively motivated dimensions:
>
>    - Learning from Instructor (LfI): (e.g., [4, 5]). Here, the key is the presence of an *interactive teaching agent*—the instructor—that can shape model behavior.
>    - Learning from Concept (LfC): (e.g., [6]). This dimension focuses on how models use declarative conceptual knowledge to generalize across tasks.
>    - Learning from Experience (LfE): (e.g., [7,8]). learning from past interactions, e.g., previous inputs, outputs, or trajectories, without explicit instruction
>
>    These categories reflect distinct **types of supervision and cognitive processing**. While not strictly orthogonal, they are **complementary** and designed to capture **different sources of learnability** (instructional, conceptual, experiential) that are often entangled in real-world settings.
>
>    **(2) Ensuring that improvements reflect learning, not mere input changes**
>
>    In our experimental design, we ensure that the only change across conditions is the **addition of content directly related to the targeted learning ability**. No other variables are introduced. The purpose is to evaluate whether models can acquire new knowledge through **instruction, concept, or experience**, and not due to unrelated input differences.
>
>    Under the same setting, **all models receive identical prompts except for the learning-related content**, allowing us to attribute performance differences solely to the intended learning mechanism.
>
>    We will clarify this distinction more explicitly in the revision. Thank you again for your advice!
>
>    ---
>
> 2. Reply to Q2 in weakness.
>
>    ---
>
>    Thank you for the comment. We agree that explaining the underlying causes is important but also highly challenging. Our focus in this work is to **identify and quantify** learning behaviors across dimensions; deeper causal analysis is left as **future work**, and we hope our framework provides a foundation for that.
>
>    ---
>
> 3. Reply to Q3 in weakness.
>
>    ---
>
>    Thank you for the valuable feedback. We agree that using a single fixed opponent (Qwen2.5-32B) may raise concerns about overfitting to its strategies.
>
>    To address this, we are currently extending our evaluation by including different Player-0:
>
>    Player0 = Phi-4
>
>    | Player1      | CH   | ST   | SB   | SM   |
>    | ------------ | ---- | ---- | ---- | ---- |
>    | gpt-4o       | 1.00 | 0.90 | 0.65 | 1.00 |
>    | gpt-4o-mini  | 0.65 | 0.88 | 0.86 | 0.72 |
>    | llama-3.1-8b | 0.65 | 0.70 | 0.50 | 0.75 |
>    | mistral-8b   | 0.40 | 0.75 | 0.28 | 0.69 |
>    | qwen2.5-7b   | 0.70 | 0.75 | 0.35 | 0.35 |
>    | qwen3-8b     | 0.75 | 0.65 | 0.50 | 0.70 |
>
>
>
>    Player0 = GPT4-o-mini
>
>    | Player1      | CH   | ST   | SB   | SM   |
>    | ------------ | ---- | ---- | ---- | ---- |
>    | gpt-4o       | 0.80 | 1.00 | 0.65 | 1.00 |
>    | gpt-4o-mini  | 0.50 | 0.55 | 0.45 | 0.55 |
>    | llama-3.1-8b | 0.25 | 0.85 | 0.10 | 0.70 |
>    | mistral-8b   | 0.15 | 0.85 | 0.20 | 0.45 |
>    | qwen2.5-7b   | 0.45 | 0.70 | 0.25 | 0.35 |
>    | qwen3-8b     | 0.40 | 0.95 | 0.55 | 0.50 |
>
>
>
>    Player0 = GPT4-o
>
>    | Player1      | CH   | ST   | SB   | SM   |
>    | ------------ | ---- | ---- | ---- | ---- |
>    | gpt-4o       | 0.60 | 0.50 | 0.45 | 0.50 |
>    | gpt-4o-mini  | 0.15 | 0.45 | 0.35 | 0.10 |
>    | llama-3.1-8b | 0.00 | 0.50 | 0.20 | 0.15 |
>    | mistral-8b   | 0.00 | 0.43 | 0.05 | 0.10 |
>    | qwen2.5-7b   | 0.25 | 0.53 | 0.05 | 0.05 |
>    | qwen3-8b     | 0.45 | 0.65 | 0.15 | 0.05 |
>
>    We agree that relying on a single Player-0 (Qwen2.5-32B) may raise concerns about overfitting to its strategies. To address this, we conducted preliminary experiments using alternative Player-0 models: **Phi-4**, **GPT-4o-mini**, and **GPT-4o**. For each Player-0, we computed the average performance of each Player-1 across four environments (CH, ST, SB, SM), ranked the models accordingly, and calculated the **Spearman rank correlation** with the original ranking under Qwen2.5-32B.
>
>    - Player-0 = **Phi-4** → ρ = **1.000**
>    - Player-0 = **GPT-4o-mini** → ρ = **0.829**
>    - Player-0 = **GPT-4o** → ρ = **0.714**
>
>    These results show that **model rankings remain largely consistent across different opponents**, suggesting that our benchmark findings are not overly sensitive to the choice of Player-0. We will include this analysis in the revised Appendix.
>
>
>
>    Regarding the evaluation metric: while win rate is a coarse signal, it has shown **strong differentiation across models** (Table 2), and aligns with our goal of evaluating whether models can learn to succeed in dynamic environments. That said, we agree that finer-grained metrics (e.g., adaptation speed, instruction adherence) could offer additional insight and are worth exploring in future versions of LearnArena.
>
>    ---
>
>
>
> 4. Reply to Q1 in Questions.
>
>    ---
>
>    In our experimental design, we ensure that the only change across conditions is the **addition of content directly related to the targeted learning ability**. No other variables are introduced. The purpose is to evaluate whether models can acquire new knowledge through **instruction, concept, or experience**, and not due to unrelated input differences.
>
>    Under the same setting, **all models receive identical prompts except for the learning-related content**, allowing us to attribute performance differences solely to the intended learning mechanism.
>
>    ---
>
> 5. Reply to Q2 in Questions.
>
>    ---
>
>    Thank you for the question. We provide a more detailed analysis of *where* the breakdown occurs in **Section 6.2** (Fig. 6): across multiple models, performance peaks and then drops as the number of in-context examples exceeds ~900. This trend is consistent even in larger models (e.g., Qwen2.5-14B), suggesting that current LLMs struggle to benefit from long trajectories in ICL.
>
>    As for *why* this degradation occurs, we believe it is a challenging open question. Possible factors include attention bottlenecks, interference effects, or insufficient abstraction, but a full explanation remains beyond the scope of this paper and is an important direction for future work.
>
>    We will clarify this point further in the revision.
>
>    ---
>
> Once again, we appreciate your valuable feedback and the opportunity to address your concerns.
>
> **Reference:**
>
> [1] The modularity of mind
>
> [2] Experiential learning: Experience as the source of learning and development
>
> [3] Putting students on the path to learning: The case for fully guided instruction
>
> [4] Rapid instructed task learning: A new window into the human brain's unique capacity for flexible cognitive control
>
> [5] Rapid instruction-based task learning (RITL) in schizophrenia
>
> [6] An Alternative to Cognitivism: Computational Phenomenology for Deep Learning
>
> [7] Experiential learning: Experience as the source of learning and development
>
> [8] ExpeL: LLM Agents Are Experiential Learners

---

> > ### Comment · Reviewer_vwyb · 2025-08-07
> >
> > Thank you for your detailed response, which adequately addresses my major concerns. I believe the additional experiments and anlaysis will strengthen the paper, and I encourage the authors to include those in the revised version. This would make the work more useful for future research and follow-up studies.
> >
> > I will adjust my score accordingly.

---

> ### Author Response · Authors · 2025-08-04
>
> Dear Reviewer vwyb,
>
> Thank you again for your valuable feedback on our submission. We hope that our responses have addressed your questions and concerns.
> If you have further question, we hope to take the last chance in the rest days to clarify.
>
> Best regards!

---

> ### Author Response · Authors · 2025-08-07
>
> Dear Reviewer vwyb,
>
> Thank you very much for your thoughtful feedback and for raising your score. We appreciate your recognition of the clarifications and additional experiments. As suggested, we will incorporate them into the revision.
>
> Thank you again for your constructive feedback throughout the review process.
>
> Best regards!

---

### Official Review · Reviewer_pKvT · 2025-07-02

**Clarity:** 2
**Significance:** 2
**Originality:** 3
**Rating:** 4
**Confidence:** 3

**Summary:**

This paper studies the learning ability of LLMs. Specifically, it proposes a framework that evaluates learning ability in three dimensions: learning from instruction, learning from concepts, and learning from experience. The authors conduct experiments to evaluate LLMs from these three perspectives and provide empirical findings about the learning abilities of LLMs. Finally, this paper introduces a benchmark, LearnArena, that evaluates the learning ability of LLMs across these dimensions.

**Questions:**

- The models' performance being worse in a many-shot setting compared to a few-shot setting could be due to their poor long-context capability. Have the authors evaluated models that are specifically designed for handling long-contexts? What's the connection between the finding here and in [1]?

[1] Rishabh Agarwal, Avi Singh, Lei M. Zhang, Bernd Bohnet, Luis Rosias, Stephanie Chan, Biao Zhang, Ankesh Anand, Zaheer Abbas, Azade Nova, John D. Co-Reyes, Eric Chu, Feryal Behbahani, Aleksandra Faust, Hugo Larochelle. Many-Shot In-Context Learning

**Ethical Concerns:**

["NO or VERY MINOR ethics concerns only"]

**Final Justification:**

Thank you to the authors for their reply! The rebuttal has addressed most of my concerns, including the clarification of the task designs and experiments on other models. I have raised my score from 3 to 4. Overall, the benchmark could be a valuable resource. I think the writing could be further improved. I also agree with the other reviewers that more discussion about the connection between this paper and the cognitive science literature would be helpful.

**Limitations:**

Yes

**Quality:**

2

**Strengths And Weaknesses:**

Strengths
- The paper studies an important direction. Understanding the learning ability of LLMs is increasingly important, and the proposed benchmark could be valuable for LLM evaluation.
- The proposed three dimensions intuitively make sense and capture different aspects of human learning.
- The experiments cover a diverse set of tasks.

Weakness
- I'm confused about the structure of the paper and the motivation for some experiments. In sections 4-6, the three different dimensions of learning ability are evaluated, but using very different settings and tasks; for example, the datasets used in section 4 are quite different from those in section 5. Is there any specific reason why this is the case? These choices should be clearly justified. As noted in Section 7, the proposed LearnArena could be used to evaluate all three dimensions. Then why not simply use this benchmark for all dimensions? It is unclear if the paper wants to focus on (1) defining the three dimensions of learning ability, (2) proposing a benchmark, or (3) just evaluating the LMs' performance. Sections 4-6 feel disconnected from section 7.
- The paper lacks a coherent conclusion about what can be learned from these empirical results. The authors list a few findings—for example, that interaction improves performance and that scaling helps—but most of these also apply to other tasks and abilities, and none are particularly surprising. It’s unclear what the main takeaway is and what direction future work should focus on for improving the learning ability of models. Also, most experiments only use Qwen models, so it’s unclear whether the same conclusions translate to other models.
- While the proposed LearnArena could be valuable, there is a lack of detail and analysis for the benchmark and the results of all evaluated models. If the paper claims a benchmark contribution, more dataset details (e.g., statistics), model performance analysis, error patterns, and qualitative analysis need to be included to better understand what the benchmark is evaluating.

---

> ### Author Rebuttal · Authors · 2025-07-31
>
> Dear Reviewer pKvT,
> Thank you so much for your valuable feedback!
>
> 1. Reply to Q1 in Weakness:
>
>    ---
>
>    Thank you for the question. Our current design follows a two-stage logic: (1) isolate and validate each learning type independently using the most suitable task setup (Sections 4–6), and (2) evaluate all three jointly under a unified benchmark (Section 7, LearnArena).
>    We intentionally avoid using a single fixed dataset (e.g., LearnArena) for all experiments, because the goal of Sections 4–6 is to independently test each learning mechanism under minimal interference from unrelated factors.
>    Specifically:
>
>    - In Section 4 (Learning from Instructor), we use math tasks (e.g., GSM8K) because they naturally support a tutor–learner setup. Math problems enable clear supervision (step-by-step reasoning), and allow us to utilize strong instructors (e.g., Qwen2.5-Math-72B) to deliver high-quality guided signals.
>    - In Section 5 (Learning from Concept), we choose tasks where injecting structured knowledge is feasible and meaningful:
>      - Competitive environments (e.g., TextArena) allow rule-based concepts to be clearly expressed.
>      - LogicGrid, NaLogic focus on formal logical constraints.
>      - Plan, AlfWorld, ScienceWorld, BabyAI test concept generalization in planning and agent settings, where task structure is well-defined.
>    - In Section 6 (Learning from Experience), we study two types of experience:
>      - On-the-fly interactions in competitive games (TextArena), where experience is generated dynamically across rounds.
>      - Pre-collected trajectories as in-context examples, used to analyze passive adaptation from past instances.
>        These choices reflect our aim to evaluate each learning type in isolation, using domains that best match the nature of the supervision (e.g., feedback, concept, or trajectory).
>        In Section 7, we introduce LearnArena as a unified benchmark that integrates all three learning dimensions in a shared game environment. The goal here is not to isolate but to holistically evaluate general learning ability under realistic multi-agent settings.
>        We will clarify this rationale in the revision.
>        Thank you again for pointing this out!
>
>    ---
>
> 2. Reply to Q2 in Weakness:
>
>    ---
>
>    Thank you for the thoughtful comment. We address “what is new” along three axes:
>
>    (1) A distinct signal from existing benchmarks.
>    Our benchmark captures learning behavior rather than static skill: system‑level Spearman correlations are low or near‑zero with GSM8K/ARC/Winogrande and only moderate with MMLU and instruction‑following sets (Fig. 10). This indicates we are measuring a capability not well covered by standard evaluations—namely, how models *learn* from instructor feedback, conceptual schemas, and prior experience.
>
>    (2) A principled way to quantify learning ability.
>    We contribute a cognitively grounded tripartite decomposition—LfI/LfC/LfE—and a relative evaluation protocol (Norm‑Acc and deltas) that isolates the effect of the learning channel from raw task skill (Secs. 4–6). We then integrate the three channels in a unified setting (Sec. 7). To our knowledge, this offers an early, reusable methodology for comparing learning ability across models, beyond absolute task scores.
>
>    (3) Non‑obvious, learning‑specific empirical findings.
>    Beyond “larger is better” or “interaction helps,” our analyses surface *where and when* learning signals help or hurt:
>
>    - Conceptual understanding is scale-emergent: structured knowledge hurts small models but helps larger ones (Sec. 5).
>      Few-shot effective, many-shot brittle: performance degrades when context becomes too long (~900 examples), highlighting limits in long-context learning (Fig. 6).
>
>    (4) On model generality: why we focus on Qwen models.
>     We use Qwen models extensively because the family includes a broad range of scales (1.5B to 72B) under a consistent architecture and training regime. This makes it uniquely well-suited for analyzing scaling effects across different learning dimensions (e.g., Fig. 2–3, 4, 5, 6).
>    That said, we also include other model families—such as GPT-4o, LLaMA3, Mistral, and Phi-3—in multiple parts of the study (e.g., Sec. 4.1, Fig. 2, Sec. 6.2, Sec. 7). These comparisons show consistent trends, supporting the robustness and generality of our conclusions beyond Qwen.
>    We agree that using a single fixed opponent (Qwen2.5-32B) may raise concerns about overfitting to its strategies.
>    To address this, we are currently extending our evaluation by including different Player-0:
>
>    ```
>    Given the rebuttal word limit, we’re happy to share the detailed table during the rebuttal phase. You may also refer to "Reply to Q3 in weakness" in Reviewer vwyb’s section for details.
>    ```
>
>    We agree that relying on a single Player-0 (Qwen2.5-32B) may raise concerns about overfitting to its strategies. To address this, we conducted preliminary experiments using alternative Player-0 models: Phi-4, GPT-4o-mini, and GPT-4o. For each Player-0, we computed the average performance of each Player-1 across four environments (CH, ST, SB, SM), ranked the models accordingly, and calculated the **Spearman rank correlation** with the original ranking under Qwen2.5-32B.
>
>    - Player-0 = **Phi-4** → ρ = 1.000
>    - Player-0 = **GPT-4o-mini** → ρ = 0.829
>    - Player-0 = **GPT-4o** → ρ = 0.714
>      These results show that **model rankings remain largely consistent across different opponents**, suggesting that our benchmark findings are not overly sensitive to the choice of Player-0. We will include this analysis in the revised Appendix.
>
>    Thank you again for the helpful suggestion!
>
>    ---
>
> 3. Reply to Q3 in Weakness:
>
>    ---
>
>    Thank you for the helpful suggestion. We agree that a benchmark claim should come with clear dataset statistics and deeper analyses. In the paper and appendix we already include several components, and we will make them more explicit and expand them in the revision:
>    **What is already included**
>
>    - Learning dynamics & many-shot analysis: Sec. 6.2 (Fig. 6) details non-monotonic ICL behavior (initial gains, peak, then decline; sharp drop >~900 examples) and capacity effects (14B > 7B), with discussion of possible interference/attention limits.
>
>    - Experience-based setup details: App. A.5 describes on-policy (multi-agent games) vs. off-policy (ICL) settings, sampling, repetition (20 runs/seeded), and evaluation protocol.
>
>    - Qualitative cases: We provide step-by-step transcripts (e.g., **Tak** example) illustrating decision trajectories:
>
>      ```
>      Given the rebuttal word limit, we’re happy to share the detailed case study during the rebuttal phase if that would be helpful.
>      ```
>
>    - **Benchmark stats (to be consolidated):** per-environment counts and lengths, e.g.:
>      Each environment contains **50 unique tasks**, with **20 runs per task**. Input lengths vary by environment, from 63.9 tokens (SpellingBee) to 839 tokens (Stratego), with the following averages: WordChains (131), SpellingBee (63.9), SpiteAndMalice (593), Tak (739), TicTacToe (166), Stratego (839), Checkers (289), and TruthAndDeception (122.9).
>
>       We will add these to the Appendix for better clarity.
>
>    ---
>
> 4. Reply to Q1 in Questions:
>
>    ---
>
>    Thank you for your question.
>
>    (a) Long-context model evaluation:
>
>    Yes, in our many-shot experiments (Section 6.2, Figure 6b), we evaluate Qwen2.5-7B/14B-Instruct-1M, which support up to 1M-token windows. These models are explicitly designed for long-context usage. We observe a clear non-monotonic trend: performance increases initially with more in-context examples, peaks (at 3–9 examples depending on model size), then declines. This reflects that LLMs remain few-shot learners, and many-shot generalization remains challenging even for long-context models.
>
>    (b) Additional evaluation with GPT-4o-mini:
>
>    Due to cost constraints (we run each experiment with 20 trials), we use **GPT-4o-mini** (128k context) instead of GPT-4o for additional verification. Following [1], we use 4-shot as a baseline and test up to 100 examples:
>
>    ICL example num → performance (GPT-4o-mini):
>
>    0: 71.2, 4: 71.4, 10: 72.8, 15: 75.2, 20: 71.2, 30: 70.8, 50: 70.0, 80: 70.6, 100: 70.6
>
>    This result reproduces the same rise-then-fall pattern reported in [1], where performance improves up to a peak and then drops.
>
>    (c) Consistency with [1]:
>
>    In [1], Section 3.1, Figure 7 (left) reports Gemini 1.5 Pro on MATH-500 with ground-truth ICL examples—a setup aligned with our many-shot configuration. Performance rises as shots increase, peaks at ~125, then declines. Although [1] truncates at 500 shots, the downward trend suggests that, if extended to >1,000 shots as in our setting—performance would continue to drop, potentially falling below zero-shot and and the 4-shot baseline used in [1]. In our experiments on MagpieMath (following the setting in Section 4) , we indeed observe the same improvement → peak → decline pattern beyond 30–100 examples, with performance eventually falling below the 4-shot baseline (noting that absolute scores differ across datasets).
>
>    Across all models—Qwen2.5-7B → Qwen2.5-14B → GPT-4o-mini → Gemini 1.5 Pro—we observe a consistent pattern:
>
>    - Performance peaks then degrades in many-shot settings.
>    - As model capability increases, the peak shifts later, and degradation happens more slowly.
>
>    This suggests that more capable models handle longer trajectories better but are still not immune to degradation.
>
>    We will include the GPT-4o-mini results and a direct comparison to [1] in the revised appendix. Thank you again for the thoughtful question—it helped us strengthen the analysis and clarify important empirical trends!
>
>    ---
>
> Once again, we appreciate your valuable feedback and the opportunity to address your concerns!
>
> **Reference:**
>
> [1]  Many-Shot In-Context Learning

---

> ### Author Response · Authors · 2025-08-04
>
> Dear Reviewer pKvT,
>
> Thank you again for your valuable feedback on our submission. We hope that our responses have addressed your questions and concerns.
> If you have further question, we hope to take the last chance in the rest days to clarify.
>
> Best regards!

---

### Official Review · Reviewer_U78M · 2025-07-03

**Clarity:** 3
**Significance:** 2
**Originality:** 2
**Rating:** 4
**Confidence:** 4

**Summary:**

Previous LLM evaluation papers focused on specific tasks, reasoning, instruction-following ability, etc., but the learning ability was rarely evaluated. This paper systematically analyzes LLMs’ learning ability from three perspectives inspired by cognitive psychology and education: learning from instructor, learning from concept, and learning from experience. The main conclusions include: (1) interaction improves learning; (2) conceptual understanding is scale-emergent and benefits larger models (for the other two perspectives, the experimental results also demonstrate that larger models are better); (3) LLMs are effective few-shot learners, but too many examples decrease performance. This paper also proposes a new benchmark, LearnArena, to systematically evaluate LLMs’ general learning ability across the three perspectives.

**Questions:**

Please refer to the weaknesses above. Besides, I have one extra question regarding understanding:

(1) What is the difference between “passive consumption” in “learning from instructor” and “experience-driven adaptation” in “learning from experience”? The “passive consumption” means “the learner receives only direct solutions from the instructor without any further interaction,” which seems similar to few-shot examples. The “experience-driven adaptation” means “LLMs learn from ICL-style demonstrations viewed as episodic traces.” It seems both are similar to few-shot examples. I am not sure whether my understanding is correct. If it is incorrect, please correct me.

**Ethical Concerns:**

["NO or VERY MINOR ethics concerns only"]

**Final Justification:**

Thank the authors for the rebuttal! The authors' response makes sense, and most of my concerns are addressed. Weaknesses 3, 4, 5, as well as my question, are addressed. For Weakness 1 (how to separate the learning ability and domain ability) and Weakness 2 (whether it is meaningful to create a new benchmark), I am not fully sure, but I agree this is a very difficult question. It is very difficult to separate the model's learning ability and domain ability (math, logic, reasoning ability, etc.) in evaluation. The authors have used a reasonable method by computing the relative gains.

Although it has some weaknesses, the paper is quite solid, it conducts extensive experiments on many datasets and models. It provides a useful new benchmark, is well written, and the motivations based on cognitive science is inspiring. The advantages outweigh the weaknesses. Therefore, I would like to change my score to borderline accept.

**Limitations:**

yes

**Quality:**

3

**Strengths And Weaknesses:**

### Strengths:

(1) It is interesting to analyze LLMs’ learning ability from the three perspectives inspired by cognitive psychology and education.

(2) The paper conducts extensive experiments on many datasets and models. The experimental results are convincing. The paper summarizes meaningful insights from the experimental results, which could be helpful for future applications.

(3) As shown in Figure 10 in the appendix, the new benchmark has different model rankings from the previous benchmarks, which demonstrates that the new benchmark is meaningful.

(4) The paper is well-written. It is easy to follow and understand. The visualizations are clear and help with my understanding.

### Weaknesses:

The paper is of high quality, and I greatly appreciate the authors’ efforts. However, I am a little concerned about the motivations.

(1) In lines 19–20, the authors stated that in previous works, “Large Language Models have demonstrated impressive capabilities across diverse tasks such as mathematics, coding, and reasoning.” However, it is difficult to evaluate LLMs’ learning ability alone. When we evaluate an LLM’s learning ability, we have to rely on some domain-specific datasets, such as math, coding, reasoning, etc., where the math/coding/reasoning ability is also involved. For example, when we compare different models’ learning ability using a math dataset, we are actually comparing their learning ability plus their math ability, instead of only the learning ability. The proposed benchmark is not evaluating the learning ability alone. I am wondering whether the methods are the best way to evaluate learning ability (although I also cannot think of better ways to evaluate learning ability alone).

(2) Following the last weakness, since many benchmark datasets already exist for math, coding, and reasoning, I am wondering whether it is meaningful to specifically evaluate learning ability and create a new benchmark for this, especially when the current setting cannot separate learning ability from math/coding/reasoning/... abilities.

(3) I am wondering why these three perspectives are chosen. There are some other perspectives related to learning, for example, learning from self-critique (we can let the LLM evaluate its own response and improve its own response). It would be better if the authors could explain more about why they chose these three perspectives.

(4) The novel insight seems limited. I think most of the paper’s conclusions are also true for LLMs’ general ability and are obvious and already well-known—for example, interaction improves performance, and the model’s capacity grows when the model is larger. It would be better if the paper could provide more novel insights or insights that are unique and specific to learning ability.

(5) It is a little unclear why the authors chose specific datasets for evaluating each perspective and why they did not use a fixed dataset, e.g., the proposed LearnArena benchmark, to analyze the three perspectives. Could you please specify the reasons behind choosing these datasets for each perspective?

---

> ### Author Rebuttal · Authors · 2025-07-31
>
> Dear Reviewer oteN,
>
> Thank you so much for your valuable feedback!
>
> 1. Reply to Q1 in weakness.
>
>    ---
>
>    Thank you for your valuable suggestion!
>
>    Indeed, evaluating learning ability independently of domain ability is fundamentally challenging, and we agree that task performance inevitably reflects both.
>
>    To mitigate this, in Sections 4–6, we **independently evaluate** each learning type, *Learning from Instructor*, *Learning from Concept*, and *Learning from Experience*, under controlled conditions using **relative performance metrics**. Specifically:
>
>    We report learning-induced improvements as **relative gains**, such as Norm Acc = training accuracy divided by original model accuracy, as shown in Figure 2 and Figure 3. Alternatively, we use delta improvements, e.g., the difference between conditions with and without concept input (see Figure 4) or score improvements after concept injection (Table 1, where darker color indicates greater gain). These analyses are designed to isolate the *effect of learning*, not task-specific competence.
>
>    In Section 7, we use environments under novel rules or new task dynamics, specifically designed to minimize reliance on models’ prior task-specific capabilities. This allows us to comprehensively assess general learning ability, integrating all three aspects, learning from instructor, concept, and experience.  We report absolute results here to allow direct comparison across test models, making the results more practically useful.
>
>    We agree your valuable suggestion. In the revision, we will add a relative gain table to LearnArena, comparing performance *before* and *after* learning. [Due to the NeurIPS rebuttal word limit, we’d be happy to share the table separately during the rebuttal phase if helpful.]
>
>    ---
>
> 2. Reply to Q2 in weakness.
>
>    ---
>
>    Thank you for the thoughtful question.
>
>    While many benchmarks exist for math, coding, and reasoning, they primarily assess static performance—what a model knows—rather than how a model learns. Our work aims to fill this gap by evaluating the ability to improve through instruction, conceptual abstraction, and experience.
>
>    Learning ability is critical for real-world applications but remains underexplored. Our benchmark provides the first systematic, cognitively grounded evaluation of this capacity.
>
>    To validate that our benchmark reflects learning ability rather than domain knowledge, we compute Spearman correlations with nine standard benchmarks (Fig. 10 in Appendix). Results show **low or near-zero correlation** with GSM8K, ARC, and Winogrande, and only moderate correlation with MMLU and instruction-following datasets. This suggests that our benchmark captures distinct behavioral capabilities, particularly around adaptation and abstraction.
>
>    In short, our work complements existing evaluations by targeting a core yet underexplored dimension of model capability: learning ability.
>
>    ---
>
> 3. Reply to Q3 in weakness.
>
>    ---
>    Thank you for raising this important point.
>
>    Our framework is grounded in long-standing perspectives from cognitive science and educational theory (e.g., [1,2,3]), which emphasize that human learning emerges through multiple complementary pathways. Drawing from these foundations, we propose a tripartite decomposition of learning ability into three cognitively motivated dimensions:
>
>    - Learning from Instructor (LfI): (e.g., [4, 5]). Here, the key is the presence of an *interactive teaching agent*—the instructor—that can shape model behavior.
>    - Learning from Concept (LfC): (e.g., [6]). This dimension focuses on how models use declarative conceptual knowledge to generalize across tasks.
>    - Learning from Experience (LfE):  (e.g., [7,8]). learning from past interactions, e.g., previous inputs, outputs, or trajectories, without explicit instruction
>
>    Regarding self-critique as an alternative learning form: we agree it is a promising direction, and we view it as a special case of Learning from Instructor, where the “instructor” is the model itself.
>
>    We appreciate the suggestion and will clarify how self-critique fits into our framework in the revision.
>
>    ---
>
> 4. Reply to Q4 in weakness.
>
>    ---
>
>    Thank you for the thoughtful comment. We address “what is new” along three axes:
>
>    **(1) A distinct signal from existing benchmarks.**
>    Our benchmark captures learning behavior rather than static skill: system‑level Spearman correlations are low or near‑zero with GSM8K/ARC/Winogrande and only **moderate** with MMLU and instruction‑following sets (Fig. 10). This indicates we are measuring a capability not well covered by standard evaluations—namely, how models *learn* from instructor feedback, conceptual schemas, and prior experience.
>
>    **(2) A principled way to quantify learning ability.**
>    We contribute a cognitively grounded tripartite decomposition—LfI/LfC/LfE—and a relative evaluation protocol (Norm‑Acc and deltas) that isolates the effect of the learning channel from raw task skill (Secs. 4–6). We then integrate the three channels in a unified setting (Sec. 7). To our knowledge, this offers an early, reusable methodology for comparing learning ability across models, beyond absolute task scores.
>
>    **(3) Non‑obvious, learning‑specific empirical findings.**
>    Beyond “larger is better” or “interaction helps,” our analyses surface *where and when* learning signals help or hurt:
>
>    - Conceptual understanding is scale-emergent: structured knowledge hurts small models but helps larger ones (Sec. 5).
>
>      Few-shot effective, many-shot brittle: performance degrades when context becomes too long (~900 examples), highlighting limits in long-context learning (Fig. 6).
>
>    ---
>
> 5. Reply to Q5 in weakness.
>
>    ---
>
>    Thank you for the question. Our current design follows a two-stage logic: (1) isolate and validate each learning type independently using the most suitable task setup (Sections 4–6), and (2) evaluate all three jointly under a unified benchmark (Section 7, LearnArena).
>
>    We intentionally avoid using a single fixed dataset (e.g., LearnArena) for all experiments, because the goal of Sections 4–6 is to independently test each learning mechanism under minimal interference from unrelated factors.
>
>    Specifically:
>
>    - In Section 4 (Learning from Instructor), we use *math tasks* because they naturally support a tutor–learner setup. Math problems enable clear supervision (step-by-step reasoning), and allow us to utilize strong instructors (e.g., Qwen2.5-Math-72B) to deliver high-quality guided signals.
>    - In Section 5 (Learning from Concept), we choose tasks where injecting structured knowledge is feasible and meaningful:
>      - *Competitive environments* (e.g., TextArena) allow rule-based concepts to be clearly expressed.
>      - *LogicGrid, NaLogic* focus on formal logical constraints.
>      - *Plan, AlfWorld, ScienceWorld, BabyAI* test concept generalization in planning and agent settings, where task structure is well-defined.
>    - In Section 6 (Learning from Experience), we study two types of experience:
>      - *On-the-fly interactions* in competitive games (TextArena), where experience is generated dynamically across rounds.
>      - *Pre-collected trajectories* as in-context examples (ICL), used to analyze passive adaptation from past instances.
>
>    These choices reflect our aim to evaluate each learning type in isolation, using domains that best match the nature of the supervision (e.g., feedback, concept, or trajectory).
>
>    In **Section 7**, we introduce LearnArena as a unified benchmark that integrates all three learning dimensions in a shared game environment. The goal here is not to isolate but to holistically evaluate general learning ability under realistic multi-agent settings.
>
>    We will clarify this rationale in the revision.
>
>    ---
>
> 6. Reply to Q1 in Questions.
>
>    ---
>
>    Rebuttal to R6 — On the distinction between passive consumption and experience-driven adaptation
>
>    Thank you for the insightful question. While both LfI and LfE may resemble few-shot inputs in format, they differ fundamentally in purpose, structure, and learning assumptions:
>
>    In passive consumption (LfI), the learner receives explicit instructional content—e.g., worked-out examples, explanations, or step-by-step reasoning—generated by a strong instructor model. The goal is didactic: to teach the model how to solve the problem, much like a teacher guiding a student through a solution. The supervision is intentional, structured, and pedagogically oriented.
>
>    In contrast, experience-driven adaptation (LfE) provides episodic traces—e.g., raw input–output pairs from prior attempts—without explicit teaching or abstraction. The learner must infer patterns, strategies, or errors on its own, resembling unsupervised or self-supervised learning. This is cognitively more demanding, as the model must decide what (if anything) is worth learning from the examples.
>
>    In short, LfI assumes the presence of instructional intent (even in passive form), whereas LfE assumes no such guidance—the model must learn from experience, not instruction. We will clarify this distinction in the revision.
>    We will include a clarification of this distinction in the Appendix, and we sincerely appreciate your helpful feedback.
>
>    ---
>
> Once again, we appreciate your valuable feedback and the opportunity to address your concerns.
>
> **Reference:**
>
> [1] The modularity of mind
>
> [2] Experience as the source of learning and development
>
> [3] Putting students on the path to learning: The case for fully guided instruction
>
> [4] Rapid instructed task learning: A new window into the human brain's unique capacity for flexible cognitive control
>
> [5] Rapid instruction-based task learning (RITL) in schizophrenia
>
> [6] An Alternative to Cognitivism: Computational Phenomenology for Deep Learning
>
> [7] Experiential learning: Experience as the source of learning and development
>
> [8] LLM Agents Are Experiential Learners

---

> > ### Comment · Reviewer_U78M · 2025-08-06
> >
> > Thank the authors for the rebuttal! The authors' response makes sense, and most of my concerns are addressed. Weaknesses 3, 4, 5, as well as my question, are addressed. For Weakness 1 (how to separate the learning ability and domain ability) and Weakness 2 (whether it is meaningful to create a new benchmark), I am not fully sure, but I agree this is a very difficult question. It is very difficult to separate the model's learning ability and domain ability (math, logic, reasoning ability, etc.) in evaluation. The authors have used a reasonable method by computing the relative gains.
> >
> > Although it has some weaknesses, the paper is quite solid, it conducts extensive experiments on many datasets and models. It provides a useful new benchmark, is well written, and the motivations based on cognitive science is inspiring. The advantages outweigh the weaknesses. Therefore, I would like to change my score to borderline accept.

---

> > > ### Author Response · Authors · 2025-08-06
> > >
> > > Dear Reviewer U78M,
> > >
> > > Thank you very much for your thoughtful feedback and for raising your score!
> > >
> > > We appreciate your recognition of our clarifications and the challenges around disentangling learning and domain ability. We're glad our approach and contributions were helpful, and we will incorporate the suggested clarifications in the revision.
> > >
> > > Best regards!

---

> ### Author Response · Authors · 2025-08-04
>
> Dear Reviewer U78M,
>
> Thank you again for your valuable feedback on our submission. We hope that our responses have addressed your questions and concerns.
> If you have further question, we hope to take the last chance in the rest days to clarify.
>
> Best regards!

---

### Note · Authors · 2025-08-16

Dear Area Chair and Reviewers,

We are writing to provide a brief summary of our rebuttal discussion. We sincerely thank you for your time and for the constructive engagement that has helped us significantly improve our work.

We are very encouraged by the positive feedback and consensus from all four reviewers after our rebuttal. Here is a summary of their individual assessments:

- We are pleased that Reviewer U78M found our paper to be solid and well-supported by extensive experiments. They indicated that our response successfully addressed most of their concerns and that our approach to the more challenging questions was reasonable
- For Reviewer pKvT, we understand that their primary concerns, especially regarding our task design and additional model experiments, have been resolved.
- Reviewer vwyb confirmed that our detailed response adequately addressed their major concerns.
- We thank Reviewer 932f for the valuable feedback on the concept scope. We are happy to adopt their suggestion to address this in the revision.

We are grateful for the clear feedback and will diligently incorporate all the valuable discussions, clarifications, and additional experiments into our revision as suggested.

In summary, as affirmed by the reviewers, our work addresses an important and unexplored gap by evaluating the dynamic learning abilities of LLMs, rather than just their static knowledge. The reviewers highlighted that our framework, inspired by cognitive science, is a valuable contribution, and they found our experiments to be extensive and the results convincing. They also agreed that our new benchmark is meaningful and provides novel insights into model behavior.

Finally, we acknowledge the limitations and valuable directions for future work identified during the review process. Our focus in this work was to identify and quantify learning behaviors across different dimensions. A deeper causal analysis to explain the underlying causes of these observed phenomena is a crucial and highly challenging next step, which we will address in future work.

Thank you once again for your time and for facilitating this constructive review process!

Best regards!

---

### Decision · Program_Chairs · 2025-09-17

**Decision:**

Accept (poster)

**Comment:**

This paper presents a framework for evaluating the learning ability of language models along three dimensions - instructions, concepts, and experience - investigates these for current models, and provides a benchmark for future evaluations. The review process was effective: reviews were helpful in identifying issues with the submitted draft, and the authors' response and promises for revisions alleviated most of the reviewers' concerns. One point that was brought up by several reviewers was the need to better clarify the connection to cognitive studies that the paper aims to establish. Another was the need for being careful about attributing conclusions to learning vs. domain ability. The authors should revise the paper to address these points and others raised by the reviewers.